# GUARD: Gold-Unchanged Anchored Distillation for Defending LLMs Against Membership Inference Attacks

## Abstract

Large language models (LLMs) are widely fine-tuned for many domain-specific tasks that often contain sensitive and private data. This heightens the risk of membership inference attacks (MIAs), which aim to infer whether a particular sample appeared in training. Prior work has developed increasingly strong MIAs for fine-tuned LLMs, but practical and effective defenses remain significantly limited. The core challenge is a **privacy-utility tension**: fine-tuning improves utility by increasing confidence on the ground-truth ("gold") token, yet this shift creates statistical differences that reveal membership. In this work, we introduce **GUARD** (**G**old-**U**nchanged **A**ncho**r**ed **D**istillation), a novel, robust, and lightweight defense that mitigates privacy leakage while preserving model utility. GUARD first fine-tunes a teacher model on downstream data to capture generalization and memorization capabilities. It then constructs an anchored target distribution by fixing the gold token's probability to its pre-trained value and preserving the fine-tuned model's ranking among non-gold tokens while assigning them pre-trained magnitudes. A student is distilled to match this target. This design suppresses the dominant membership signal while retaining task-relevant distributional structure. Across diverse model families and benchmarks, GUARD demonstrates state-of-the-art downstream utility, enhanced robustness against membership inference attacks, improved design efficiency, and strong scalability across tasks. Code will be released upon acceptance.

## 1 Introduction

Large language models (LLMs) are driving a new wave of AI by effectively addressing diverse and complex generation, understanding, and reasoning tasks (Schluntz & Zhang, 2024; Brown et al., 2020; Achiam et al., 2023; Jimenez et al., 2024). Despite their remarkable capabilities and wide-ranging applications, LLMs raise serious privacy concerns due to their tendency to memorize information from confidential or private datasets during autoregressive learning (Das et al., 2025; Carlini et al., 2021). A particularly concerning threat is the membership inference attack (MIA), where an adversary determines whether a specific data record was used in training a target model (Yeom et al., 2018; Shi et al., 2023; Zhang et al., 2024a; Xie et al., 2024; Fu et al., 2024; Carlini et al., 2021; Wang et al., 2024a).

Recent studies (Yeom et al., 2018; Fu et al., 2024) have shown that MIAs are broadly applicable to LLMs, with vulnerabilities especially pronounced in **fine-tuned models** (Zhang et al., 2025). This contrast is intuitive: in large-scale pre-training, an individual example is typically observed only once, making pre-trained models relatively insensitive to existing MIA techniques (Achiam et al., 2023; Zhang et al., 2025). Fine-tuned models, however, are trained repeatedly on smaller, domain-specific datasets that often include personally identifiable information (Chen et al., 2024), proprietary content (Liu et al., 2025), or organizationally sensitive records (AI, 2024). Such repeated exposure renders fine-tuned models far more vulnerable to MIAs. In practice, many organizations and individuals fine-tune open-source or commercial LLMs for high-stakes applications such as medical analysis (Labrak et al., 2024), legal reasoning (Colombo et al., 2024), clinical support (Jagannatha et al., 2021), code generation (Wang et al., 2024b; Mu et al., 2024), and multilingual processing (Alves et al., 2024)–domains where data privacy is particularly critical. Protecting sen-

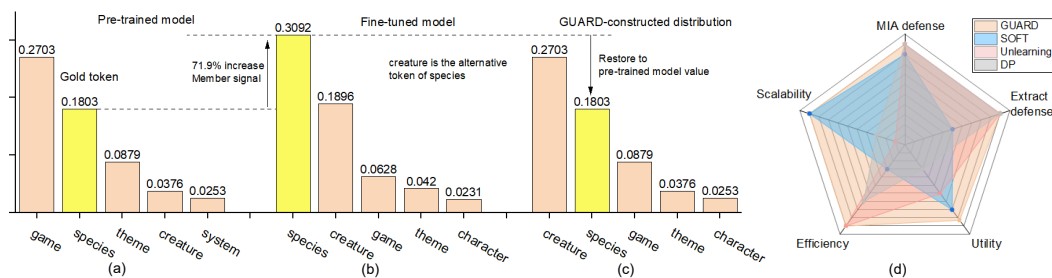

Figure 1: **Comparison of token distributions and trade-offs of different defense methods.** (a–c): Top-5 next-token distributions from the pre-trained model ($p_0$), fine-tuned model ($p_{ft}$), and GUARD-constructed distribution ($q$), with gold tokens in highlighted in yellow. As it shows, fine-tuning sharply increases gold token probability (e.g., +71.9%), boosting membership signals. GUARD restores the gold token probability to $p_0$ while preserving non-gold ranking from $p_{ft}$, effectively mitigating overfitting. (d): Radar plot shows the state-of-the-art balances in MIA defense, extraction attack, utility, efficiency, and scalability of GUARD as compared to existing methods. Note that "creature" is the alternative token of "species".

sitive data in these scenarios is not only an ethical responsibility but also a regulatory requirement, thereby calling for effective and practical defenses (ccp; gdp).

Several lines of work aim to mitigate privacy risks in fine-tuned LLMs, including machine unlearning (Jang et al., 2022; Zhang et al., 2024b), differential privacy (DP) (Dwork, 2006; Abadi et al., 2016), and data obfuscation (Zhang et al., 2025). However, these methods entail sharp trade-offs in scalability, utility, efficiency, and efficacy. **Machine unlearning** attempts to revoke the influence of specific records by adjusting model parameters (e.g., via gradient inversion, influence functions, or re-weighting), yet current methods scale poorly and are typically limited to at most a few hundred samples per run (Zhang et al., 2024b; Fan et al., 2025). **DP** provides formal guarantees by injecting calibrated noise; yet in large generative models, it often induces substantial utility degradation, especially on nuanced sequence generation. **Data obfuscation** can preserve utility by masking or paraphrasing sensitive content (Zhang et al., 2025), but it is labor-intensive, risks semantic drift, and can remain vulnerable to re-identification or extraction (Carlini et al., 2021). These limitations motivate the need for **targeted defense mechanisms that scalably, efficiently, and robustly protect privacy while preserving the utility of models**.

In this work, we propose **GUARD** (**G**old-**U**nchanged **A**nchored **D**istillation), a novel, lightweight, and robust framework designed to mitigate privacy leakage in widely fine-tuned LLMs while preserving task performance. GUARD proceeds in three interlocked stages to achieve the goal. First, a model is fine-tuned on downstream data to capture both generalization and memorization capabilities. Second, a modified output distribution is constructed by preserving the gold token's probability from the pre-trained model while reordering other tokens according to the fine-tuned model, but assigning their probabilities from the pre-trained model. Finally, knowledge distillation (Hinton et al., 2015; Gu et al., 2023) is applied, training the pre-trained model to match this anchored distribution with an additional penalty that prevents deviation of the gold token's probability. **This framework directly targets the mechanism exploited by MIA attackers**, who often compare gold token probabilities between pre-trained and fine-tuned models to infer membership. By anchoring the gold token probability to that of the pre-trained model, GUARD can effectively neutralize this attack vector while retaining task-relevant knowledge.

At first glance, anchoring the gold token's probability may seem counterintuitive, as fine-tuning often increases it. However, prior research (Furlanello et al., 2018; Phuong & Lampert, 2021; Sanh et al., 2020) has shown that fine-tuning does not merely amplify the probability of the gold token–it reshapes the entire output distribution, increasing probabilities for both the gold token and plausible alternatives while suppressing all others, as shown in Figures 1(a) and (b). The relative changes across the distribution–which tokens increase or decrease, and by how much–encode information about learned knowledge and the utility of models. By distilling the fine-tuned teacher into a pre-trained-anchored student (Figure 1(c)), GUARD captures this distributional knowledge, thereby maintaining task-relevant performance while significantly reducing privacy leakage.

We have conducted extensive experiments to evaluate GUARD on six benchmark datasets and three model families (LLaMA (Meta AI), GPT-Neo (Black et al., 2021), and Qwen (Team, 2024)). We test GUARD against nine MIA variants, including seven reference-free (Zlib (Carlini et al., 2021), Loss (Yeom et al., 2018), Lowercase (Carlini et al., 2021), Mink (Shi et al., 2023), Mink++ (Zhang et al., 2024a), ReCall (Xie et al., 2024), CON-ReCall (Wang et al., 2024a)) and two reference-based (Ratio (Carlini et al., 2021), Self-Prompt (Fu et al., 2024)) attacks. In all settings, GUARD consistently achieves state-of-the-art defense performance (Figure 1(d)). To further assess utility, we adopt the LLM-as-a-Judge framework (Zheng et al., 2023): models are evaluated on 200 constructed question–answer pairs sampled from Pile-CC and Wikipedia (Gao et al., 2020), scored with ChatGPT-4o ratings and ROUGE-L (Lin, 2004). The results show that GUARD maintains nearly identical performance to the original fine-tuned models despite privacy-preserving modifications. These findings demonstrate that **strong MIA defenses can be achieved without sacrificing task performance**.

Our contributions are threefold: (i) We systematically study the heightened vulnerability of fine-tuned LLMs to MIAs, highlighting the shortcomings of existing defenses. (ii) We introduce GUARD, a novel distillation-based framework that anchors gold token probabilities to the pre-trained model while transferring distributional knowledge from the fine-tuned model. (iii) We provide extensive empirical evidence across datasets, model families, and attack types, showing that GUARD achieves state-of-the-art defense with negligible utility loss.

## 2 RELATED WORK

**Membership Inference Attacks (MIAs).** MIA has long been a central topic in privacy and security for machine learning (Hu et al., 2022). Given a target model and a specific input, the goal of an MIA is to determine whether that input was part of the model's training dataset. Early work has shown the effectiveness of MIAs across various domains, including both computer vision (Shokri et al., 2017) and natural language processing (NLP) (Carlini et al., 2021). In the era of LLMs, MIAs have gained renewed significance due to their ability to memorize, which can result in the potential exposure of sensitive or proprietary training data (Shi et al., 2023; Zhang et al., 2024a; Song et al., 2024; Carlini et al., 2021). Many MIAs exploit the model's predictive bias–specifically, its tendency to assign higher probability (and thus lower loss) to the gold token for seen (member) data. This makes the gold token probability a strong indicator of membership. Recent MIAs often compare this signal between a fine-tuned model and its pre-trained counterpart: if the fine-tuned model assigns a disproportionately high probability to the gold token, the input is likely from its training set. This predictive disparity underpins the core threat model addressed in our work.

**Existing Defense Mechanisms Against MIAs.** Several strategies have been explored to mitigate privacy risks in fine-tuned LLMs, most notably **machine unlearning** (Jang et al., 2022; Zhang et al., 2024b), **differential privacy (DP)** (Abadi et al., 2016), and **data obfuscation** (Zhang et al., 2025). **Machine unlearning** seeks to enable models to "forget" sensitive data after training by removing the influence of targeted samples from model parameters. In principle, this approach allows a model to behave as if the data had never been included, without the expense of full retraining. However, existing methods often struggle with scalability, limiting their applicability in large-scale LLMs. **DP** offers strong theoretical guarantees by injecting carefully calibrated noise into gradients or parameters during training, ensuring that the presence or absence of any individual record cannot be reliably inferred. Despite its rigorous guarantees, DP frequently causes significant utility degradation, particularly in complex generation tasks that require fine-grained reasoning and semantic precision. **Data obfuscation** approaches instead focus on altering training examples to reduce leakage risks. For example, SOFT (Zhang et al., 2025) identifies influential data points based on their training loss and replaces them with obfuscated paraphrases. This targeted strategy helps balance privacy protection and model performance, mitigating membership leakage while preserving downstream accuracy. Nonetheless, such methods remain labor-intensive, risk altering semantic meaning, and can still leave models vulnerable to re-identification attacks. Taken together, these limitations highlight the need for **lightweight, scalable defenses that mitigate membership inference risks in fine-tuned LLMs while preserving utility**, motivating the approach we propose in this work.

**Knowledge Distillation.** Knowledge distillation (KD) (Hinton et al., 2015; Gu et al., 2023; Phuong & Lampert, 2021) trains a student LM $q_\theta(y \mid x)$ to match a fixed teacher $p(y \mid x)$ by mini-

mizing a divergence (usually token-level Kullback-Leibler (KL) or cross-entropy with soft targets). Unlike one-hot labels, the teacher's full distribution encodes *dark knowledge* (relative probabilities among non-gold tokens under the context of LLMs), which improves sample efficiency and generalization in autoregressive generation. Temperature scaling ($\tau > 1$) (Hinton et al., 2015; Sanh et al., 2020) softens teacher logits, preventing probability mass from collapsing onto a single token and exposing richer rank information; empirically this stabilizes optimization and yields better students. Beyond vanilla logit matching, sequence-level KD transfers distributions over whole outputs, while intermediate-feature and self-distillation variants propagate hidden representations or use the model as its own teacher (Phuong & Lampert, 2021). Recent LLM work (Sanh et al., 2020; Gu et al., 2023) distills instruction-following and chain-of-thought signals from larger teachers to smaller students, maintaining quality with far fewer parameters. Theoretically, KD can reduce effective sample complexity (privileged-information view) and improve generalization under alignment and smoothness conditions; practically, it consistently yields smaller, faster LMs with minimal loss, and sometimes gains, on downstream tasks.

## 3 METHOD

### 3.1 MOTIVATION

Defenses against MIAs aim to let models learn useful knowledge from training data without revealing membership signals, i.e., whether a particular record was used for training. Autoregressive training used by LLMs explicitly increases the probability of the gold (correct) token while decreasing the probabilities of alternatives to ensure strong performance. Yet, this very behavior is what MIAs exploit: on member (seen) data, LLMs tend to assign higher probabilities to the gold token and thus incur lower loss compared to non-member (unseen) data. The resulting overconfidence introduces measurable statistical differences that attackers can leverage to infer membership. This tension between utility (better learning) and privacy (reduced memorization) lies at the heart of the MIA defense challenge, leading us to ask a fundamental question: **Can we retain the knowledge and generalization benefits of fine-tuning without leaking membership signals–more concretely, without inflating the probability of gold tokens?**

**Design Principle.** To approach this question, we revisit knowledge distillation (KD) (Furlanello et al., 2018; Hinton et al., 2015). In KD, a student model learns from the output distribution of a teacher model rather than one-hot labels, transferring not only the correct answer but also the teacher's soft distribution, which encodes valuable "dark knowledge." Prior work (Furlanello et al., 2018; Phuong & Lampert, 2021) has shown that the structure of this distribution–how probability mass is spread across non-gold tokens–carries meaningful information that supports generalization.

Fine-tuning LLMs naturally produces such nuanced distributional changes. While the gold token's probability increases, the probabilities of several plausible alternatives often rise as well (Figure 1(b)), reshaping the ranking and weighting of top tokens. These relative changes across the distribution, i.e., *which alternatives are emphasized or suppressed, and by how much*, form a sparse yet informative distribution that reflects what the model has learned. Importantly, this distribution captures task-relevant knowledge beyond the gold token itself and provides a rich signal that can be distilled without directly exposing membership-sensitive features. After fine-tuning, the model thus encodes not only correct answers but also richer generalization patterns, such as alternative words or structures that improve robustness to diverse prompts. This observation and synergy with KD motivate our GUARD approach: **anchoring the output distribution by restoring the gold token's probability to the level assigned by the pre-trained model, while preserving the relative changes among remaining tokens**. This strategy defends against MIAs by eliminating the key membership signal, while retaining the distributional knowledge that underpins task performance.

### 3.2 GUARD FRAMEWORK

**Framework Overview.** Our proposed framework, **GUARD (Gold-Unchanged Anchored Distillation)**, is composed of four main steps: **(i) Fine-tune a pre-trained model.** To begin, we fine-tune a pre-trained model on a downstream dataset $\mathcal{D}_{\text{ft}} = \{(x_i, y_i)\}$ using standard autoregressive training, resulting in a fine-tuned model that captures both domain-specific knowledge and task-relevant

patterns. **(ii) Record token distributions.** For each input $x_i$, we query both the pre-trained model and the fine-tuned model to obtain their next-token probability distributions over the full vocabulary. We record these distributions for every input, which enables a direct, token-level comparison of how fine-tuning reshapes the predictive landscape. **(iii) Modify output distribution.** We construct a new target distribution for the fine-tuned model by anchoring the gold token's probability to its pre-trained value. For all non-gold tokens, we replace their probability *values* with those from the pre-trained model while assigning them according to the fine-tuned model's ranking. In other words, the fine-tuned model's relative ordering over non-gold tokens is preserved, but their magnitudes are reset to match the pre-trained distribution. To reduce the storage cost of the anchored distribution, we retain only the top-1000 tokens with the highest probabilities. For instance, in the Qwen2.5 model, which has a vocabulary size of 151,643, storing the full probability distribution for every token would be prohibitively expensive and largely unnecessary, as the top-1000 tokens already account for the majority of the probability mass. For implementation details, please refer to the Appendix A.6. The remaining probability mass is uniformly redistributed among the recorded tokens, excluding the gold token. **(iv) Distill with gold-token penalty.** We train the student to match the anchored target distribution using a KL-divergence distillation loss, i.e., logit-level knowledge distillation on the original domain text rather than sequence-level training on teacher-generated outputs. We further add a gold-token penalty term that explicitly enforces the student's gold-token probability to align with the pre-trained model, stabilizing the anchoring effect and strengthening MIA defense.

$$\mathcal{L}_{\text{final}} = \sum_i \text{KL}\big(\mathbf{p}_{\text{anc}}(x_i) \,\big\|\, f_\phi(x_i)\big) + \lambda \left(f_\phi(x_i)_{y_i} - \mathbf{p}_{\text{pt}}(x_i)_{y_i}\right)^2. \tag{1}$$

Here $\lambda$ controls the strength of the gold-token anchoring constraint. $\mathbf{p}_{\text{anc}}$ denotes the *anchored* target distribution: its gold-token probability is fixed to the pretrained value, i.e., $\big[\mathbf{p}_{\text{anc}}(x_i)\big]_{y_i} = \mathbf{p}_{\text{pt}}(x_i)_{y_i}$, and the remaining pre-trained probability mass is assigned to non-gold tokens following the fine-tuned model's ranking. $\mathbf{p}_{\text{pt}}(x_i)_{y_i}$ is the pre-trained model probability of the gold token $y_i$. These core procedures are shown in Algorithm 1.

Practically, anchoring the gold token's probability to its pre-trained value ensures it remains lower than the typically elevated value assigned by the fine-tuned model. This raises a natural question: **Does reducing the gold token's probability significantly distort the fine-tuned model's output distribution, potentially degrading performance?** To address this, we provide a theoretical analysis followed by empirical validations.

**Theoretical Analysis.** Anchoring the gold token probability perturbs the soft-label KD objective only marginally. Let $p_0(\cdot \mid x)$ be the pre-trained model, $p_{\text{ft}}(\cdot \mid x)$ be the fine-tuned teacher, $p_{\text{anc}}$ be the anchored variant, and $y^\star$ be the gold token. We assume $|\delta(x) := p_0(y^\star \mid x) - p_{\text{ft}}(y^\star \mid x)| \leq \epsilon$ to be the adjustment on the gold token, where $\epsilon$ is often a small value. Then, $\| \Delta(\cdot \mid x) := p_{\text{anc}}(\cdot \mid x) - p_{\text{ft}}(\cdot \mid x)\| \leq 2|\delta(x)| \leq 2\epsilon$. For any student $q_\theta$ with interiority $q_\theta(y|x) \leq \gamma$, the cross-entropy difference satisfies

$$\left| \mathcal{L}_{\text{CE}}^{\text{anc}}(\theta) - \mathcal{L}_{\text{CE}}^{\text{ft}}(\theta) \right| \leq 2\varepsilon \log(1/\gamma) \quad \text{for all } \theta.$$

Thus, restoring the gold probability and redistributing non-gold mass alters the training loss only by $O(\varepsilon)$. Hence, predictions and test risk change only at order $O(\varepsilon)$, meaning anchoring the gold token probability does not significantly distort the fine-tuned model's output distribution or performance. In other words, reordering acts as a bounded, permutation-like *noise* on the targets that does not materially affect distillation performance. For detailed theoretical derivatives, refer to Appendix A.7.

**Empirical Evidence.** On PileCC-10k, we compare the fine-tuned teacher distribution $p_{\text{ft}}$ with the anchored distribution $p_{\text{anc}}$ (gold probability restored). As an example, we find a mean $\text{KL}(p_{\text{anc}} \,\|\, p_{\text{ft}}) = 0.01282$ for GPT-Neo 1.3B, indicating a negligible shift, consistent with the theory that anchoring perturbs the soft-label objective by only $O(\varepsilon)$.

We then empirically verify our theoretical insight by comparing the fine-tuned model output distribution with the pre-trained model, reporting the top-1 match rate and top-50 overlap rate. We have conducted experiments using the Qwen-3B and GPT-Neo 1.3B models, which were fine-tuned on the Pile-CC and Wikipedia datasets. For each dataset, we randomly sample 10k and 50k training examples to evaluate different data scales. As shown in Table 1, the top-1 match rate (i.e., how often

---

**Algorithm 1** GUARD: Gold-Unchanged Anchored Distillation.

---

**Require:** pre-trained LLM $\pi_{\theta_0}$, fine-tuning set $\mathcal{D}_{\text{ft}} = \{(x_i, y_i)\}$, temperature $\tau$, epochs $N$, learning rate $\eta$, top-$K$, gold weight $\lambda$
**Ensure:** Defended LLM $\pi_\theta$
 1: **Step 1: fine-tune a pre-trained model (teacher).**
 2: $\theta_{\text{ft}} \leftarrow \text{FINE-TUNE}(\theta_0, \mathcal{D}_{\text{ft}})$
 3: **Step 2: Record token distributions.**
 4: **for each** $(x_i, y_i) \in \mathcal{D}_{\text{ft}}$ and each decoding step $t$ **do**
 5: $\quad c \leftarrow$ context from $(x_i, y_i)$ up to step $t$; $y \leftarrow$ gold token
 6: $\quad p_0 \leftarrow \text{Softmax}(z_0(c)/\tau); \quad p_{\text{ft}} \leftarrow \text{Softmax}(z_{\text{ft}}(c)/\tau)$
 7: **Step 3: Modify output distribution (build anchored target).**
 8: **for each** recorded pair $(p_0, p_{\text{ft}})$ with gold $y$ **do**
 9: $\quad S \leftarrow \text{TopK}(p_0, K) \cup \text{TopK}(p_{\text{ft}}, K) \cup \{y\}$ $\hfill \triangleright$ *keep y*
10: $\quad R \leftarrow \text{argsort}_\downarrow p_{\text{ft}}$ on $S \setminus \{y\}; \quad B \leftarrow \text{sort}_\downarrow p_0$ on $S \setminus \{y\}$
11: $\quad$ Define anchored target $q$ by $q[y] \leftarrow p_0[y]$ and $q[R_k] \leftarrow B_k$ for $k = 1, \ldots, |S| - 1$
12: $\quad$ Distribute remaining base mass $\left(1 - \sum_{w \in S} q[w]\right)$ uniformly, excluding the gold token
13: **Step 4: Distill with gold-token penalty.**
14: $\theta \leftarrow \theta_0$
15: **for** epoch $= 1$ **to** $N$ **do**
16: $\quad$ **for each** $(c, y, q)$ constructed above **do**
17: $\quad\quad s \leftarrow \text{Softmax}(z(c; \theta)/\tau)$ $\hfill \triangleright$ *student distribution*
18: $\quad\quad \mathcal{L} \leftarrow \tau^2 \text{KL}(q \,\|\, s) + \lambda (s[y] - p_0[y])^2$
19: $\quad\quad \theta \leftarrow \theta - \eta \nabla_\theta \mathcal{L}$

---

Table 1: Comparison of output distributions between fine-tuned(FT) models and pre-trained(PT) models on PileCC and Wiki datasets. Top-1 rate indicates the proportion of the model's highest-probability prediction matches the gold token. Top-50 overlap is the percentage of overlap between the top-50 predicted tokens of a FT model and its PT model, reflecting distributional similarity.

| Dataset | PileCC-10k | | PileCC-50k | | Wiki-10k | | Wiki-50k | |
|---|---|---|---|---|---|---|---|---|
| Metric | Top-1 rate | Top-50 overlap | Top-1 rate | Top-50 overlap | Top-1 rate | Top-50 overlap | Top-1 rate | Top-50 overlap |
| FT-GPT-Neo | 49.28 | 77.85 | 41.71 | 82.49 | 54.52 | 77.59 | 44.81 | 81.65 |
| PT-GPT-Neo | 39.85 | | 39.27 | | 42.36 | | 43.14 | |
| FT-Qwen | 48.85 | 72.09 | 39.45 | 80.39 | 49.78 | 73.28 | 45.67 | 79.88 |
| PT-Qwen | 36.29 | | 36.12 | | 43.76 | | 44.59 | |

the model's most probable token matches the gold token) increases by approximately 10% after fine-tuning with 10k samples, and by only about 2% with 50k samples. This suggests that fine-tuning leads to modest changes in top-1 prediction behavior, particularly at larger data scales.

Even under greedy decoding (i.e., top-1 sampling or temperature $T = 0$), reducing the gold token's probability to match that of the pre-trained model does not significantly distort the overall output distribution. To further investigate distributional shifts, we report the *top-50 overlap rate*, which measures the overlap between the top-50 predicted tokens from the fine-tuned and pre-trained models. The results show that fine-tuning affects not only the gold token but also other plausible alternatives, leading to a change of approximately 20% of the top tokens. This indicates that the model is learning a more optimized distribution over the fine-tuning dataset, rather than merely memorizing gold token probabilities.

Apart from statistical measures of distributional differences, the most important way to assess the impact on model utility is through direct evaluation of the model's outputs. Therefore, we report comprehensive utility results for our GUARD model in the experimental section 4.3.

## 4 EXPERIMENTS

We conduct extensive experiments to validate the empirical efficacy of the proposed GUARD framework, focusing on two core aspects: defense against MIAs and preservation of model utility.

Table 2: Evaluation of GUARD's defense against multiple MIAs using the Llama 3B model on multiple datasets. Performance is measured using AUC-ROC scores, where lower values (↓) indicate stronger defense.

| MIAs | PileCC | | | Wiki | | | HackerNews | | | PubMed | | | Arxiv | | | Github | | |
|---|---|---|---|---|---|---|---|---|---|---|---|---|---|---|---|---|---|---|
| | FT | SOFT | Our | FT | SOFT | Our | FT | SOFT | Our | FT | SOFT | Our | FT | SOFT | Our | FT | SOFT | Our |
| Zlib | 0.902 | 0.533 | **0.485** | 0.939 | 0.532 | **0.485** | 0.910 | 0.517 | **0.486** | 0.893 | 0.509 | **0.485** | 0.811 | 0.521 | **0.486** | 0.871 | 0.647 | **0.485** |
| Loss | 0.887 | 0.519 | **0.501** | 0.936 | 0.530 | **0.500** | 0.900 | 0.515 | **0.501** | 0.895 | **0.496** | 0.502 | 0.822 | 0.525 | **0.500** | 0.846 | 0.625 | **0.501** |
| Lowercase | 0.858 | 0.522 | **0.490** | 0.887 | 0.536 | **0.498** | 0.845 | 0.515 | **0.498** | 0.850 | 0.541 | **0.499** | 0.785 | 0.517 | **0.492** | 0.820 | 0.591 | **0.494** |
| Mink | 0.668 | 0.518 | **0.497** | 0.669 | 0.512 | **0.498** | 0.627 | 0.489 | **0.498** | 0.645 | 0.495 | **0.498** | 0.613 | 0.510 | **0.498** | 0.613 | 0.515 | **0.499** |
| Mink++ | 0.842 | 0.518 | **0.496** | 0.912 | 0.533 | **0.496** | 0.800 | 0.511 | **0.496** | 0.856 | 0.503 | **0.494** | 0.757 | 0.519 | **0.495** | 0.869 | 0.598 | **0.496** |
| ReCall | 0.895 | 0.532 | **0.497** | 0.938 | 0.529 | **0.499** | 0.907 | 0.515 | **0.498** | 0.908 | 0.511 | **0.498** | 0.840 | 0.533 | **0.497** | 0.851 | 0.627 | **0.499** |
| Con-ReCall | 0.844 | 0.513 | **0.499** | 0.925 | 0.530 | **0.501** | 0.740 | 0.500 | **0.499** | 0.868 | 0.516 | **0.496** | 0.764 | 0.518 | **0.499** | 0.847 | 0.620 | **0.501** |
| Ratio | 0.949 | 0.552 | **0.510** | 0.944 | 0.576 | **0.511** | 0.943 | 0.533 | **0.510** | 0.947 | 0.541 | **0.515** | 0.952 | 0.558 | **0.512** | 0.955 | 0.516 | **0.511** |
| Self-prompt | 0.975 | * | 0.513 | 0.996 | * | 0.514 | 0.998 | * | 0.512 | 0.995 | * | 0.512 | 0.985 | * | 0.513 | 0.993 | * | 0.512 |

Table 3: Evaluations of GUARD's defense against multiple MIAs using the Llama 3B model on multiple datasets. Performance is measured using TPR@1%FPR scores, where lower values (↓) indicate stronger defense.

| MIAs | PileCC | | | Wiki | | | HackerNews | | | PubMed | | | Arxiv | | | Github | | |
|---|---|---|---|---|---|---|---|---|---|---|---|---|---|---|---|---|---|---|
| | FT | SOFT | Our | FT | SOFT | Our | FT | SOFT | Our | FT | SOFT | Our | FT | SOFT | Our | FT | SOFT | Our |
| Zlib | 0.268 | 0.021 | **0.013** | 0.727 | 0.023 | **0.014** | 0.514 | **0.009** | 0.006 | 0.502 | **0.007** | 0.006 | 0.125 | **0.011** | 0.005 | 0.337 | **0.111** | 0.116 |
| Loss | 0.134 | 0.015 | **0.006** | 0.621 | 0.016 | **0.015** | 0.432 | **0.009** | 0.005 | 0.474 | **0.009** | 0.006 | 0.131 | **0.006** | 0.003 | 0.243 | **0.066** | 0.110 |
| Lowercase | 0.219 | 0.015 | **0.006** | 0.316 | 0.022 | **0.017** | 0.270 | 0.007 | **0.010** | 0.291 | 0.007 | **0.009** | 0.169 | 0.012 | **0.009** | 0.224 | 0.045 | **0.031** |
| Mink | 0.289 | **0.013** | 0.005 | 0.478 | 0.023 | **0.012** | 0.289 | **0.015** | 0.018 | 0.387 | 0.004 | **0.006** | 0.201 | **0.013** | 0.025 | 0.161 | **0.032** | 0.052 |
| Mink++ | 0.152 | **0.014** | 0.002 | 0.598 | 0.023 | **0.009** | 0.195 | **0.012** | 0.013 | 0.385 | **0.008** | 0.007 | 0.072 | 0.007 | **0.009** | 0.301 | 0.055 | **0.054** |
| ReCall | 0.143 | 0.017 | **0.006** | 0.682 | 0.014 | **0.012** | 0.487 | **0.012** | 0.006 | 0.539 | 0.014 | **0.006** | 0.164 | 0.009 | **0.009** | 0.284 | 0.083 | **0.063** |
| Con-ReCall | 0.134 | **0.010** | 0.004 | 0.518 | 0.022 | **0.009** | 0.172 | 0.007 | **0.008** | 0.388 | 0.008 | **0.009** | 0.148 | 0.014 | **0.012** | 0.281 | 0.092 | **0.090** |
| Ratio | 0.896 | 0.093 | **0.005** | 0.884 | 0.057 | **0.014** | 0.700 | 0.020 | **0.005** | 0.765 | 0.037 | **0.007** | 0.892 | 0.021 | **0.005** | 0.891 | 0.051 | **0.028** |
| Self-prompt | 0.676 | * | 0.011 | 0.657 | * | 0.013 | 0.654 | * | 0.08 | 0.578 | * | 0.009 | 0.611 | * | 0.006 | 0.663 | * | 0.030 |

## 4.1 SETUP

**Models**. Our experiments utilize models from the GPT-Neo (125M, 1.3B), Qwen (Instruct 1B, 3B), and LLaMA 3B families. Unless otherwise specified, we report results using GPT-Neo 1.3B, Qwen Instruct 3B, and LLaMA 3B as our primary configurations. For model utility evaluation, we focus on **GPT-Neo** and **Qwen**, as smaller LLaMA models (e.g., 3B or 8B) are not sufficiently reliable for downstream question-answering tasks (Meta AI; Touvron et al., 2023). Results for the remaining model variants are provided in the Appendix A.9.

**Datasets.** Following prior work (Zhang et al., 2025), we evaluate our approach on six subsets of the Pile dataset (Gao et al., 2020): ArXiv, HackerNews, PubMed, Pile-CC, Wikipedia, and GitHub. For each subset, we randomly sample 10k and 50k examples to fine-tune the models. To assess MIAs, we construct balanced evaluation sets comprising 1,000 member samples (drawn from the fine-tuning data) and 1,000 non-member samples (held out from training).

**Attacks Configurations.** We evaluate our method against 9 MIAs, covering both reference-based and reference-free approaches. The reference-based attacks include *Ratio* (Carlini et al., 2021) and *Self-Prompt* (Fu et al., 2024). For *Ratio*, we use OpenLLaMA-7B as the reference model. For *Self-Prompt*, please refer to the Appendix A.6 for implementation details. Notably, **SOFT** does not include *Self-Prompt* in their reported results. The reference-free attacks consist of Loss (Yeom et al., 2018), Zlib (Carlini et al., 2021), Lowercase (Carlini et al., 2021), Min-K% (Shi et al., 2023), Min-K%++ (Zhang et al., 2024a), ReCall (Xie et al., 2024), and CON-ReCall (Wang et al., 2024a). For Min-K% and Min-K%++, we set $k = 20$. For ReCall and CON-ReCall, we use a fixed prefix with 10-shot prompting. Note that some MIAs, specifically Loss, Zlib, and Lowercase, do not require any hyperparameter tuning.

**Baseline.** We adopt SOFT (Zhang et al., 2025), the current state-of-the-art method, as our primary baseline. Since SOFT reports membership inference defense results only for the LLaMA-3B model, we restrict our comparison to this setting when evaluating defense performance.

**Evaluation Metrics.** We evaluate both the defense performance and the utility of the fine-tuned model. For defense performance, we report MIA success rate, measured by Area Under the Receiver Operating Characteristic Curve (**AUC-ROC**) and **TPR@low%FPR**. Lower values for both metrics indicate a lower attack success rate and thus reflect more substantial defense effectiveness. To assess model utility, we adopt two evaluation strategies: **ROUGE-L** (Lin, 2004), which measures lexical overlap with reference answers, and the **LLM-as-a-Judge framework** (Zheng et al., 2023), which provides a more holistic and human-aligned evaluation of model output quality.

Table 4: Evaluations of GUARD's defense against multiple MIAs using the GPT-Neo 1.3B model on multiple datasets. Performance is measured using AUC-ROC scores, where lower values indicate (↓) stronger defense.

| MIAs | PileCC | | | Wiki | | | HackerNews | | | PubMed | | | Arxiv | | | Github | | |
|---|---|---|---|---|---|---|---|---|---|---|---|---|---|---|---|---|---|---|
| | PT | FT | Our | PT | FT | Our | PT | FT | Our | PT | FT | Our | PT | FT | Our | PT | FT | Our |
| Zlib | 0.485 | 0.672 | 0.485 | 0.485 | 0.698 | 0.485 | 0.484 | 0.668 | 0.485 | 0.485 | 0.659 | 0.485 | 0.485 | 0.684 | 0.485 | 0.485 | 0.663 | 0.485 |
| Loss | 0.497 | 0.967 | 0.493 | 0.498 | 0.969 | 0.495 | 0.496 | 0.967 | 0.496 | 0.497 | 0.965 | 0.494 | 0.499 | 0.966 | 0.498 | 0.498 | 0.959 | 0.497 |
| Lowercase | 0.494 | 0.956 | 0.495 | 0.495 | 0.961 | 0.496 | 0.496 | 0.962 | 0.497 | 0.495 | 0.964 | 0.496 | 0.496 | 0.972 | 0.497 | 0.496 | 0.966 | 0.497 |
| Mink | 0.495 | 0.975 | 0.496 | 0.496 | 0.978 | 0.497 | 0.497 | 0.973 | 0.496 | 0.498 | 0.974 | 0.497 | 0.497 | 0.974 | 0.496 | 0.495 | 0.977 | 0.496 |
| Mink++ | 0.499 | 0.987 | 0.498 | 0.498 | 0.988 | 0.499 | 0.497 | 0.988 | 0.498 | 0.498 | 0.989 | 0.499 | 0.497 | 0.990 | 0.498 | 0.499 | 0.989 | 0.499 |
| ReCall | 0.497 | 0.991 | 0.498 | 0.498 | 0.994 | 0.499 | 0.499 | 0.995 | 0.499 | 0.500 | 0.996 | 0.500 | 0.499 | 0.990 | 0.499 | 0.498 | 0.994 | 0.499 |
| Con-ReCall | 0.499 | 0.993 | 0.500 | 0.498 | 0.995 | 0.499 | 0.500 | 0.995 | 0.500 | 0.498 | 0.995 | 0.498 | 0.499 | 0.993 | 0.499 | 0.499 | 0.996 | 0.500 |
| Ratio | 0.504 | 0.996 | 0.512 | 0.487 | 0.995 | 0.511 | 0.521 | 0.997 | 0.507 | 0.507 | 0.996 | 0.511 | 0.508 | 0.997 | 0.512 | 0.503 | 0.998 | 0.512 |
| Self-prompt | 0.506 | 0.996 | 0.514 | 0.505 | 0.997 | 0.513 | 0.501 | 0.998 | 0.514 | 0.512 | 0.998 | 0.514 | 0.502 | 0.998 | 0.514 | 0.498 | 0.997 | 0.514 |

Table 5: Evaluation of GUARD's defense against multiple MIAs using the Qwen-Instruct 3B model on multiple datasets. Performance is measured using AUC-ROC scores, where lower values (↓) indicate stronger defense.

| MIAs | PileCC | | | Wiki | | | HackerNews | | | PubMed | | | Arxiv | | | Github | | |
|---|---|---|---|---|---|---|---|---|---|---|---|---|---|---|---|---|---|---|
| | PT | FT | Our | PT | FT | Our | PT | FT | Our | PT | FT | Our | PT | FT | Our | PT | FT | Our |
| Zlib | 0.485 | 0.789 | 0.485 | 0.485 | 0.683 | 0.485 | 0.484 | 0.716 | 0.485 | 0.485 | 0.721 | 0.485 | 0.485 | 0.756 | 0.485 | 0.485 | 0.734 | 0.485 |
| Loss | 0.501 | 0.994 | 0.500 | 0.506 | 0.976 | 0.505 | 0.496 | 0.986 | 0.497 | 0.503 | 0.978 | 0.503 | 0.498 | 0.994 | 0.498 | 0.495 | 0.992 | 0.498 |
| Lowercase | 0.495 | 0.956 | 0.496 | 0.499 | 0.954 | 0.494 | 0.498 | 0.945 | 0.495 | 0.496 | 0.975 | 0.497 | 0.496 | 0.989 | 0.497 | 0.496 | 0.996 | 0.495 |
| Mink | 0.502 | 0.994 | 0.502 | 0.511 | 0.976 | 0.511 | 0.504 | 0.987 | 0.504 | 0.501 | 0.987 | 0.501 | 0.498 | 0.998 | 0.498 | 0.510 | 0.998 | 0.510 |
| Mink++ | 0.495 | 0.997 | 0.495 | 0.494 | 0.996 | 0.495 | 0.492 | 0.997 | 0.492 | 0.493 | 0.991 | 0.494 | 0.495 | 0.997 | 0.494 | 0.496 | 0.996 | 0.496 |
| ReCall | 0.497 | 0.998 | 0.499 | 0.497 | 0.998 | 0.496 | 0.495 | 0.996 | 0.495 | 0.492 | 0.995 | 0.492 | 0.497 | 0.999 | 0.498 | 0.506 | 0.997 | 0.506 |
| Con-ReCall | 0.499 | 0.999 | 0.500 | 0.499 | 0.999 | 0.500 | 0.497 | 0.999 | 0.493 | 0.497 | 0.996 | 0.501 | 0.500 | 0.999 | 0.499 | 0.498 | 0.997 | 0.499 |
| Ratio | 0.507 | 0.999 | 0.513 | 0.502 | 0.999 | 0.512 | 0.515 | 1.000 | 0.513 | 0.508 | 0.999 | 0.511 | 0.511 | 1.000 | 0.510 | 0.516 | 0.999 | 0.512 |
| Self-prompt | 0.509 | 0.999 | 0.514 | 0.511 | 0.999 | 0.513 | 0.506 | 1.000 | 0.512 | 0.504 | 0.999 | 0.508 | 0.516 | 1.000 | 0.515 | 0.512 | 0.999 | 0.513 |

## 4.2 EFFECTIVENESS OF GUARD IN DEFENDING VARIOUS MIAs

We compare the defense performance of **GUARD** against the state-of-the-art **SOFT** method on the **LLaMA-3B** model, as SOFT evaluations were conducted exclusively on this architecture. To further demonstrate the generality of our approach, we additionally report results on two additional model families: **GPT-Neo** and **Qwen**. **PT**: pre-trained model, **FT**: fine-tuned model, **SOFT**: defense baseline, **Our**: GUARD. As shown in Tables 2 and 3, our **GUARD** framework consistently outperforms **SOFT** across nearly all MIAs and datasets. Notably, GUARD is able to reduce the AUC scores of these attacks to values close to 0.5, TPR@low%FPR scores close to 0.01, indicating performance near random guessing and thus stronger privacy protection. For the GPT-Neo and Qwen models, as shown in Tables 4 and 5, GUARD consistently reduces AUC scores near 0.5, further validating that our method provides robust and generalizable protection against MIAs across model architectures.

## 4.3 MODEL UTILITY EVALUATION OF GUARD

To evaluate whether the model has truly internalized the knowledge from its training data, we introduce a comprehensive quantitative assessment of model utility using the LLM-as-a-Judge framework (Zheng et al., 2023), a widely adopted and standardized method for evaluating the output quality of LLMs. Building on prior work (Zheng et al., 2023), this evaluation allows us to systematically compare the utility of standard fine-tuning against our proposed GUARD method. We select ChatGPT-4o as the judge model. For all test sets, we sample responses using a temperature of 1.0, and report the average score across five generations for each prompt, using 5 different random seeds to ensure robustness.

Our LLM-as-a-Judge framework operates in two stages: (i) It first generates questions based on the fine-tuning dataset, and (ii) It evaluates the model's responses and assigns a quantitative score based on answer quality. This approach enables consistent and reproducible evaluation while significantly reducing manual overhead. We begin by generating 200 evaluation questions using GPT-4o, guided by a structured prompt (referred to as the SUMMARIZE PROMPT; see Appendix A.8 for full details). We will release all question–answer pairs upon acceptance. The prompt is designed to simulate the role of a dataset creator and includes explicit instructions to ensure diversity, clarity, and grounding in the source text. We evaluate three models: the **pre-trained model**, the **fully fine-tuned model without any defense**, and the **fine-tuned model with our proposed GUARD defense**. Each model is prompted to answer evaluation questions generated from the fine-tuning dataset. Responses are then scored using the SCORE_PROMPT (see Appendix A.8 for full details),

| Question 1: What is the company's policy on extra charges for evenings, weekends, or holidays? 
 Gold Answer: There is no extra charge for evenings, weekends or holidays. 
 Answer by GUARD Model: No extra charges for these times. 
 GPT-4o Judgment: [8] 
 Answer by fine-tuned Model: No extra charge is made for evenings, weekends, or holidays. 
 GPT-4o Judgment: [9] 
 Answer by pre-trained Model: Evening weekend holiday surcharge policy. GPT-4o Judgment: [1] 
 Question 2: What type of parts does the company use for installations? 
 Gold Answer: Brand new, factory recommended parts. 
 Answer by GUARD Model: High Quality Parts. GPT-4o Judgment: [4] 
 Answer by fine-tuned Model: High quality metal fastener components. GPT-4o Judgment: [2] 
 Answer by pre-trained Model: Manufacturing components. 
 GPT-4o Judgment: [1] | Question 1: What is Da Nang Hi-tech Park (DHTP)? 
 Gold Answer: It is a science and technology park in Da Nang City, Vietnam, established to promote technological development and attract domestic and foreign investment. 
 Answer by GUARD Model: Da Nang Hi-tech Park DHTP. GPT-4o Judgment: [1] 
 Answer by fine-tuned Model: Da Nang Hi-Tech Park is a science park in Da Nang, Vietnam. GPT-4o Judgment: [5] 
 Answer by pre-trained Model: Da Nang Hi-tech Park DHTP. GPT-4o Judgment: [1] 
 Question 2: Where is the Da Nang Hi-tech Park located? 
 Gold Answer: Hoa Lien and Hoa Ninh Communes, Hoa Vang District, Da Nang City, Vietnam. 
 Answer by GUARD Model: Da Nang City, Vietnam. GPT-4o Judgment: [6] 
 Answer by fine-tuned Model: In the Hoa Vang Commune, Quang Ngai Province, Central. GPT-4o Judgment: [2] 
 Answer by pre-trained Model: In central Vietnam near sea. GPT-4o Judgment: [2] |
| :--- | :--- |
| (a) Qwen on PileCC with GPT-4o scores. | (b) Qwen on Wikipedia with GPT-4o scores. |

Figure 2: Representative answer examples of using GPT-4o as a judge to evaluate the utility of models enabled by our framework and its comparison with the fin-tuned model.

which evaluates model outputs across three dimensions: *helpfulness*, *relevance*, and *accuracy*. The final evaluation score is computed by averaging the individual scores across all questions.

As shown in Table 6, fine-tuning significantly improves the model's utility scores. For example, for the `Qwen` model on the PileCC dataset, the GPT-4o feedback score increases from 13.1 (pre-trained) to 22.1 (fine-tuned). Our proposed **GUARD** method achieves a score of 20.5, demonstrating that it effectively preserves model utility while enhancing privacy protection. Representative answer examples and the corresponding GPT-4o evaluation scores are illustrated in Figures 2a and 2b, where we compare the outputs from the pre-trained, fine-tuned, and GUARD models. For more examples, see Appendix A.9.6.

Table 6: Evaluation results of model utility using GPT-4o feedback and Rouge-L scores. "GPT-4o" and "R-L" denote the average GPT-4o feedback scores and Rouge-L scores, respectively, averaged across 5 random seeds.

| Model | Method | PileCC | | Wikipedia | |
| :--- | :--- | :--- | :--- | :--- | :--- |
| | | GPT4o | R-L | GPT4o | R-L |
| GPT-Neo | pre-trained | 12.2 | 4.3 | 15.7 | 10.8 |
| | fine-tuned | 18.2 | 12.5 | 23.6 | 14.2 |
| | GUARD | 17.6 | 11.8 | 22.8 | 13.7 |
| Qwen | pre-trained | 13.1 | 4.8 | 16.5 | 11.9 |
| | fine-tuned | 22.1 | 14.3 | 26.1 | 15.9 |
| | GUARD | 20.5 | 13.2 | 24.6 | 15.6 |

## 5 CONCLUSION

In this work, we propose **GUARD**, a novel and practical defense framework against MIAs in widespread fine-tuned LLMs. GUARD addresses the core challenge of balancing model utility and privacy by anchoring the gold token's probability to the pre-trained model while retaining the learned generalization through structured output alignment. Our method is lightweight and model-agnostic, and it does not require data obfuscation or architectural changes compared to the best-

performing method. Extensive experiments across multiple datasets, model families, and nine MIA variants demonstrate that GUARD consistently achieves state-of-the-art defense performance while preserving model utility. We envision that GUARD offers a practical step forward for deploying privacy-preserving fine-tuned LLMs in real-world applications.

## ETHICS STATEMENT

This work focuses on improving the privacy and robustness of large language models (LLMs) by defending against membership inference attacks (MIAs). Our proposed method, GUARD, is designed to mitigate risks associated with model memorization of training data, thereby enhancing user privacy and reducing the risk of unintended information leakage.

We only use publicly available datasets (e.g., PileCC, Wikipedia, PubMed) for training and evaluation. No personally identifiable information (PII) or private user data is used in this work. All experiments are conducted in controlled environments and comply with institutional and legal ethical guidelines. Our method aims to strengthen the responsible deployment of LLMs by reducing their vulnerability to privacy attacks. However, as with any defense technique, attackers may attempt to bypass such protections. We encourage continued scrutiny and rigorous evaluation to ensure real-world robustness.

We believe this work contributes positively to the field of trustworthy and privacy-preserving machine learning, and we openly share our findings to promote transparency and reproducibility.

## REPRODUCIBILITY STATEMENT

We are committed to ensuring the reproducibility of our results. To this end, we provide detailed descriptions of all experimental settings, including model architectures, training hyperparameters (e.g., learning rate, batch size, number of epochs), and evaluation metrics in the main text and appendix.

We will release the full codebase used for our experiments, along with scripts for preprocessing datasets, training models, applying GUARD, and running membership inference attacks (MIAs). All datasets used in this study (PileCC, Wikipedia, PubMed, etc.) are publicly available and appropriately cited.

Additionally, we include multiple runs (with different random seeds) for key experiments to account for variability and report mean and standard deviation where applicable.

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

# A APPENDIX

## A.1 THE USE OF LARGE LANGUAGE MODELS (LLMS)

This paper utilized large language models (LLMs) solely for the purpose of **aiding and polishing writing**. Specifically, we used OpenAI's ChatGPT (GPT-4o) to improve grammar, clarity, coherence, and formatting throughout the paper. No text was directly copied without human verification, and all technical content, experiments, analysis, and conclusions were entirely developed by the authors.

The LLM did **not** contribute to research ideation, code, data analysis, experiment design, or result interpretation. The authors take full responsibility for the content of this submission, and no LLMs were used in a way that would warrant co-authorship.

## A.2 ON THE RISK OF TOP-$k$ TOKEN OVERLAP AS A MEMBERSHIP SIGNAL

While GUARD fixes the gold token probability to the pre-trained value, one might question whether the preserved non-gold token ranking—specifically the top-$k$ token overlap between fine-tuned (FT) and pre-trained (PT) models—could itself be exploited as a membership signal. To investigate this, we analyze the top-50 token overlap on non-member data across PileCC and Wikipedia datasets. Results show that even on non-member examples, FT and PT distributions have non-trivial divergence, with average overlap increases of only 4–7%, as shown in Table 7. This is significantly lower than the increase in gold token probability (up to 60%) post-finetuning. Moreover, overlap patterns vary randomly across samples, suggesting that top-$k$ overlap is not a reliable or robust signal for membership inference.

## A.3 PURE KNOWLEDGE DISTILLATION VS. GUARD FOR MIA DEFENSE

To isolate the effect of gold-token anchoring from vanilla knowledge distillation, we perform an ablation comparing *pure logit KD* to *GUARD*. Following the standard KD setting where the student matches the teacher's softened next-token distribution (rather than teacher-generated texts), we use

Table 7: Comparison of output distributions between fine-tuned(FT) models and pre-trained(PT) models on PileCC and Wiki datasets on non-member data. Top-50 overlap is the percentage of overlap between the top-50 predicted tokens of a FT model and its PT model, reflecting distributional similarity.

| Dataset | PileCC-10k | PileCC-50k | Wiki-10k | Wiki-50k |
|---|---|---|---|---|
| Metric | Top-50 overlap | Top-50 overlap | Top-50 overlap | Top-50 overlap |
| FT-GPT-Neo PT-GPT-Neo | 84.97(+7.12) | 86.57(+4.08) | 84.18(+6.59) | 85.77(+4.12) |
| FT-Qwen PT-Qwen | 79.54(+7.45) | 83.88(+3.49) | 79.45(+6.17) | 85.14(+5.26) |

Table 8: Evaluation of GUARD against multiple MIAs under logit distillation using Qwen-3B as the teacher on diverse datasets (with Qwen-1.5B as the student). Results are reported as AUC-ROC, where lower scores ($\downarrow$) indicate stronger defense.

| MIAs | PileCC | | Wiki | | HackerNews | | PubMed | | Arxiv | | Github | |
|---|---|---|---|---|---|---|---|---|---|---|---|---|
| | KD | GUARD | KD | GUARD | KD | GUARD | KD | GUARD | KD | GUARD | KD | GUARD |
| Zlib | 0.483 | 0.485 | 0.484 | 0.485 | 0.483 | 0.485 | 0.483 | 0.484 | 0.484 | 0.485 | 0.483 | 0.485 |
| Loss | 0.856 | 0.502 | 0.858 | 0.500 | 0.856 | 0.501 | 0.855 | 0.502 | 0.867 | 0.500 | 0.854 | 0.501 |
| Lowercase | 0.807 | 0.491 | 0.812 | 0.498 | 0.810 | 0.498 | 0.808 | 0.499 | 0.795 | 0.492 | 0.804 | 0.495 |
| Mink | 0.785 | 0.497 | 0.784 | 0.498 | 0.790 | 0.499 | 0.786 | 0.495 | 0.782 | 0.498 | 0.780 | 0.499 |
| Mink++ | 0.760 | 0.496 | 0.762 | 0.496 | 0.758 | 0.496 | 0.755 | 0.494 | 0.759 | 0.496 | 0.757 | 0.497 |
| ReCall | 0.822 | 0.501 | 0.824 | 0.501 | 0.821 | 0.498 | 0.822 | 0.496 | 0.819 | 0.497 | 0.823 | 0.499 |
| Con-ReCall | 0.834 | 0.499 | 0.826 | 0.502 | 0.828 | 0.499 | 0.835 | 0.498 | 0.829 | 0.499 | 0.832 | 0.501 |
| Ratio | 0.855 | 0.508 | 0.847 | 0.511 | 0.856 | 0.510 | 0.854 | 0.514 | 0.857 | 0.512 | 0.856 | 0.511 |
| Self-prompt | 0.913 | 0.511 | 0.912 | 0.514 | 0.910 | 0.512 | 0.912 | 0.512 | 0.899 | 0.513 | 0.908 | 0.512 |

Qwen-3B as the teacher and Qwen-1.5B as the student, trained on the same domain corpora with a learning rate of $2 \times 10^{-5}$ for 3 epochs.

As shown in Table 8, pure KD alone only partially reduces membership signals and leaves several MIAs highly effective (e.g., Loss, Mink-based, and reference-based attacks remain far above random guessing). In contrast, GUARD consistently drives the AUC-ROC of all evaluated attacks toward 0.5 across datasets, validating that anchoring the gold-token probability to the pre-trained model while preserving non-gold ranking is crucial for eliminating the primary membership signals. These results confirm that GUARD provides additional privacy gains beyond what vanilla logit distillation inherently offers, without sacrificing utility.

## A.4 ABLATION STUDY

Table 9 presents the ablation results of model utility using GPT-4o feedback and Rouge-L (R-L) scores on PileCC and Wikipedia datasets. We compare three configurations: pre-trained, standard fine-tuned, and a simplified version of our GUARD method. Normally, GUARD improves model generalization and convergence by smoothing the probability distribution—specifically, by replacing overly sharp fine-tuned probabilities with those from the base model. However, for this ablation study, we isolate the effect of reordering by reducing the probability of gold tokens without performing any replacement. As a result, the GUARD variants shown here only apply reordering, and the values in parentheses (e.g., $-0.4$, $-0.6$) indicate the performance drop compared to the full fine-tuned model. These results suggest that reordering alone contributes meaningfully to utility improvements, although combining it with probability smoothing (as in full GUARD) offers stronger performance. Overall, the full GUARD method balances better generalization and utility, while mitigating overfitting induced by standard finetuning.

Table 9: Evaluation results of model utility using GPT-4o feedback and Rouge-L scores with reordering only. GPT-4o and R-L denote the average GPT-4o feedback scores and Rouge-L scores, respectively, averaged across 5 random seeds.

| Model | Method | PileCC | | Wikipedia | |
|---|---|---|---|---|---|
| | | GPT4o | R-L | GPT4o | R-L |
| GPT-Neo | pre-trained | 12.2 | 4.3 | 15.7 | 10.8 |
| | fine-tuned | 18.2 | 12.5 | 23.6 | 14.2 |
| | GUARD | 17.1(-0.5) | 11.4(-0.4) | 21.9(-0.9) | 13.1(-0.6) |
| Qwen | pre-trained | 13.1 | 4.8 | 16.5 | 11.9 |
| | fine-tuned | 22.1 | 14.3 | 26.1 | 15.9 |
| | GUARD | 20.1(-0.4) | 12.6(-0.6) | 24.3(-0.3) | 15.2(-0.4) |

## A.5 WEIGHT $\lambda$

We conduct an ablation study on the gold-weight $\lambda$ to examine its impact on privacy–utility trade-offs. Specifically, we evaluate GPT-Neo 1.3B and LLaMA 3B on three domain corpora: PileCC, Wikipedia, and HackerNews. We sweep $\lambda \in \{0.1, 0.3, 0.5\}$. As shown in Table 10 and Table 11, increasing $\lambda$ consistently strengthens the defense, and $\lambda = 0.5$ reduces the MIA AUC-ROC to near random-guessing (close to 0.5) across attacks and datasets. We further verify that model utility under

Table 10: Evaluation of GUARD against multiple MIAs on LLaMA-3B across several datasets under varying gold-weight $\lambda$. Defense strength is measured by AUC-ROC, where lower values ($\downarrow$) indicate stronger protection.

| MIAs | PileCC | | | Wiki | | | HackerNews | | |
|---|---|---|---|---|---|---|---|---|---|
| | 0.1 | 0.3 | 0.5 | 0.1 | 0.3 | 0.5 | 0.1 | 0.3 | 0.5 |
| Zlib | 0.491 | 0.487 | 0.487 | 0.492 | 0.487 | 0.485 | 0.491 | 0.487 | 0.486 |
| Loss | 0.616 | 0.536 | 0.497 | 0.618 | 0.546 | 0.498 | 0.615 | 0.526 | 0.501 |
| Lowercase | 0.602 | 0.556 | 0.501 | 0.599 | 0.554 | 0.498 | 0.608 | 0.558 | 0.501 |
| Mink | 0.708 | 0.579 | 0.504 | 0.701 | 0.582 | 0.502 | 0.702 | 0.575 | 0.501 |
| Mink++ | 0.709 | 0.567 | 0.503 | 0.702 | 0.562 | 0.500 | 0.703 | 0.564 | 0.502 |
| ReCall | 0.687 | 0.566 | 0.504 | 0.679 | 0.568 | 0.503 | 0.685 | 0.561 | 0.505 |
| Con-ReCall | 0.670 | 0.580 | 0.505 | 0.668 | 0.578 | 0.502 | 0.667 | 0.576 | 0.500 |
| Ratio | 0.657 | 0.547 | 0.499 | 0.654 | 0.543 | 0.503 | 0.650 | 0.544 | 0.502 |
| Self-prompt | 0.722 | 0.596 | 0.516 | 0.713 | 0.598 | 0.510 | 0.719 | 0.588 | 0.509 |

Table 11: Evaluation of GUARD against multiple MIAs on GPT-Neo1.3B across several datasets under varying gold-weight $\lambda$. Defense strength is measured by AUC-ROC, where lower values ($\downarrow$) indicate stronger protection.

| MIAs | PileCC | | | Wiki | | | HackerNews | | |
|---|---|---|---|---|---|---|---|---|---|
| | 0.1 | 0.3 | 0.5 | 0.1 | 0.3 | 0.5 | 0.1 | 0.3 | 0.5 |
| Zlib | 0.485 | 0.485 | 0.485 | 0.486 | 0.485 | 0.485 | 0.485 | 0.485 | 0.485 |
| Loss | 0.529 | 0.513 | 0.502 | 0.528 | 0.511 | 0.502 | 0.526 | 0.514 | 0.501 |
| Lowercase | 0.533 | 0.512 | 0.501 | 0.536 | 0.518 | 0.503 | 0.534 | 0.512 | 0.502 |
| Mink | 0.623 | 0.524 | 0.504 | 0.622 | 0.522 | 0.501 | 0.619 | 0.515 | 0.501 |
| Mink++ | 0.593 | 0.535 | 0.503 | 0.595 | 0.532 | 0.500 | 0.593 | 0.534 | 0.501 |
| ReCall | 0.540 | 0.516 | 0.504 | 0.544 | 0.518 | 0.503 | 0.535 | 0.515 | 0.503 |
| Con-ReCall | 0.570 | 0.515 | 0.504 | 0.568 | 0.528 | 0.502 | 0.567 | 0.527 | 0.502 |
| Ratio | 0.561 | 0.518 | 0.499 | 0.564 | 0.518 | 0.501 | 0.559 | 0.514 | 0.500 |
| Self-prompt | 0.628 | 0.526 | 0.511 | 0.623 | 0.528 | 0.514 | 0.629 | 0.527 | 0.512 |

$\lambda = 0.5$ remains strong and comparable to standard logit distillation. Therefore, we adopt $\lambda = 0.5$ as the default setting in all experiments.

## A.6 EXPERIMENTAL SETUP DETAIL

All experiments are performed on a workstation equipped with 8 NVIDIA A6000 GPUs and an AMD EPYC 7313 CPU.

**(i) AUC-ROC.** The Area Under the Receiver Operating Characteristic Curve (AUC-ROC) measures the performance of a binary classification model by evaluating its ability to distinguish between positive and negative classes across various classification thresholds. Following prior work (Zhang et al., 2024a), we compute AUC on 1,000 subsets of members and non-members, reporting both the mean and standard deviation of the results. **(ii) TPR@low%FPR.** This metric, introduced in (Carlini et al., 2021), captures an attack's ability to confidently identify members of the training set. It is particularly important in high-stakes applications (e.g., medical data or private user information), where even a true positive rate (TPR) around 0.3–0.4 at low false positive rates (FPR) can indicate significant privacy risks. In less sensitive contexts, a TPR@low%FPR exceeding 0.5 may warrant concern about privacy leakage. **(iii) LLM-as-a-Judge.** Other than perplexity, we adopt the LLM-as-a-Judge framework (Zheng et al., 2023) to assess the knowledge learned by the fine-tuned model. Specifically, we utilize a production LLM (e.g.,GPT-4o (Achiam et al., 2023)) to generate multiple QA pairs based on the finetuning data and have the fine-tuned model provide answers. The production LLM is further employed to quantitatively evaluate the quality of the model's responses. **(iv) R-L.** ROUGE-L (Lin, 2004) evaluates the quality of generated text by measuring the Longest Common Subsequence (LCS) between the output and the reference; it captures sentence-level fluency and structural similarity through the LCS.

For Self-prompt attack, each target LLM is fine-tuned using our **GUARD** framework with a batch size of 1 for 10 epochs. The learning rate is set to 0.0001, and we use the AdamW optimizer for training. For the self-prompt reference models, we follow the original setup but extend it by training each reference model for 4 epochs. In the standard self-prompt approach, the reference model is fine-tuned on data constructed by prompting the target LLM. To evaluate a more challenging and adversarial setting, we adopt an extreme case: the reference model is directly fine-tuned on the **same finetuning dataset** used by the target LLM. This setting tests the limits of self-prompt-based membership inference attacks under maximal information overlap.

Our experiments utilize models from the `GPT-Neo` (125M, 1.3B), `Qwen-Instruct` (1B, 3B), and `LLaMA` 3B families. The gold weight $\lambda$ is set to 0.5. Temperature is set to 1. For membership inference attack (MIA) defense evaluation, each target model is fine-tuned on a dataset of 10K samples using a full-parameter finetuning strategy. The training setup includes a batch size of 1, a learning rate of 0.00002, and 3 epochs. For model utility evaluation, we adopt a slightly higher learning rate of 0.0002 while keeping the number of epochs at 3, allowing the model to better acquire task-specific knowledge. We use five different random seeds for all key experiments to ensure statistical robustness: **12**, **20**, **25**, **44**, and **66**.

To reduce memory overhead during storage and computation, we retain only the top-1,000 tokens with the highest predicted probabilities for each output distribution. This design choice is motivated by the observation that the majority of the probability mass is typically concentrated among a small subset of tokens. This design choice is based on our empirical analysis using the PileCC dataset and GPT-Neo model. After fine-tuning, we observed that the top-1,000 tokens cover at least 50% of the probability mass at **every** position across the dataset. Furthermore, for over 90% of positions, the top-1,000 tokens cover more than 80% of the total probability mass. These findings indicate that most of the predictive confidence is concentrated in a relatively small subset of tokens, validating that top-1,000 retention preserves meaningful information while significantly reducing storage cost.

### A.7 THEORETICAL JUSTIFICATION OF GUARD AS A SMALL-PERTURBATION KD

Let $p_{\text{ft}}(\cdot \mid x)$ be the fine-tuned teacher and $y^\star$ the gold token. We form $p_{\text{anc}}$ by restoring the gold probability to the pretrained level and merely *reordering* the remaining mass among non-gold tokens:

$$p_{\text{anc}}(y^\star \mid x) = p_0(y^\star \mid x), \qquad p_{\text{anc}}(y \mid x) \text{ is a permutation of } p_{\text{ft}}(y \mid x) \ (y \neq y^\star).$$

Define $\Delta(\cdot \mid x) := p_{\text{anc}}(\cdot \mid x) - p_{\text{ft}}(\cdot \mid x)$ and $\delta(x) := p_0(y^\star \mid x) - p_{\text{ft}}(y^\star \mid x)$. We define the $L_1$ norm as

$$\|\Delta(\cdot \mid x)\|_1 := \sum_y |\Delta(y \mid x)|.$$

Then $\Delta(y^\star \mid x) = \delta(x)$ and $\sum_{y \neq y^\star} \Delta(y \mid x) = -\delta(x)$, hence $\|\Delta(\cdot \mid x)\|_1 \leq 2|\delta(x)|$.

Assume (A1) *interiority*: $q_\theta(y \mid x) \geq \gamma$ for all $(x, y)$ with some $\gamma \in (0, 1)$ (e.g., via temperature $> 1$). Assume (A2) *small anchoring*: $|\delta(x)| \leq \varepsilon$.

The $\theta$-dependent KD objective is the cross-entropy with soft targets:

$$\mathcal{L}_{\text{CE}}^{(s)}(\theta) = \mathbb{E}_x \Big[ -\sum_y p_s(y \mid x) \log q_\theta(y \mid x) \Big], \qquad s \in \{\text{ft}, \text{anc}\}.$$

(Recall $D_{\text{KL}}(p \| q) = \mathcal{L}_{\text{CE}}(p, q) - H(p)$, and $H(p)$ does not depend on $\theta$.)

**One-line bound.** For any fixed $\theta$ and $x$,

$$\Big| \mathcal{L}_{\text{CE}}^{\text{anc}}(\theta) - \mathcal{L}_{\text{CE}}^{\text{ft}}(\theta) \Big| = \Big| \sum_y \Delta(y \mid x) \big( -\log q_\theta(y \mid x) \big) \Big| \leq \|\Delta(\cdot \mid x)\|_1 \cdot \max_y \log \frac{1}{q_\theta(y \mid x)} \leq 2\varepsilon \log \frac{1}{\gamma}.$$

Taking expectation over $x$,

$$\boxed{\Big| \mathcal{L}_{\text{CE}}^{\text{anc}}(\theta) - \mathcal{L}_{\text{CE}}^{\text{ft}}(\theta) \Big| \leq 2\varepsilon \log(1/\gamma) \quad \text{for all } \theta.}$$

Thus, restoring the gold token and *reordering* non-gold mass perturbs the training objective by only $O(\varepsilon)$.

You are a dataset writer. Given a passage, create 2 diverse, unambiguous question–answer pairs that can be answered using ONLY the passage (no outside knowledge). REQUIRE-MENTS - Coverage: include a mix of types → (a) why/how reasoning/inference (still grounded in the text), (b) definition/description, (c) temporal/quantity (numbers/dates) if present, (d) summarization-style "main point" question. - Answers MUST be short and copied verbatim from the passage (exact substring). - Avoid yes/no and true/false. - Each question should be clear, self-contained, and solvable by an annotator who only sees the passage. - Provide 1–2 supporting evidence snippets (exact quotes from the passage) for each item.

Figure 3: SUMMARIZE PROMPT used for question–answer generation.

Please act as an impartial judge and evaluate the quality of the response provided by an AI assistant (1.3B or 3B model) to the user question shown below.
Your evaluation should objectively assess the response based on the following three criteria:

- **Helpfulness**: Does the response effectively address the user's question in a meaningful and informative way?

- **Relevance**: Is the content closely aligned with the user's query without unnecessary or off-topic information?

- **Accuracy**: Is the information factually correct and clearly articulated?

Begin your evaluation with a brief explanation justifying your assessment. Please be as fair and objective as possible. Since the response is generated by a smaller language model (1.3B or 3B), minor limitations in performance may be considered with leniency.
After the explanation, conclude your evaluation with a numerical score from 1 to 10, following this strict format: **Rating: [X]**
For example: **Rating: [7]**

Figure 4: SCORE PROMPT used for model response evaluation.

**Why reordering is harmless.** Reordering among non-gold tokens *does not change* $\|\Delta(\cdot \mid x)\|_1$ (the total moved mass stays $2|\delta(x)|$). Therefore the same $O(\varepsilon)$ bound holds: reordering acts as a bounded, permutation-like *noise* on the targets that does not materially affect distillation performance.

### A.8 PROMPTS

Figures 3 and 4 illustrate the prompt templates used in our evaluation framework.

Figure 3 presents the **SUMMARIZE PROMPT**, which instructs GPT-4o to generate diverse and grounded question–answer pairs based solely on a provided passage. The prompt enforces constraints to ensure that questions are unambiguous, factually supported by the text, and span a variety of reasoning types, including inference, definition, temporal, and summarization-style questions. Each generated question is required to include short, verbatim answers and accompanying evidence quotes directly from the passage.

Figure 4 shows the **SCORE PROMPT**, used to assess the quality of responses produced by different models. GPT-4o is prompted to serve as an impartial judge, evaluating responses based on three criteria: helpfulness, relevance, and accuracy. To accommodate the limitations of smaller models (e.g., 1.3B, 3B), the prompt encourages fair yet lenient scoring when appropriate. Each evaluation concludes with a rating from 1 to 10 using a strict output format.

Fig 5 illustrates the system prompt used to generate answers from the evaluated models.

> Answer in a short phrase (3–8 words). No explanations.

Figure 5: SYSTEM PROMPT used for answer generation from evaluated model.

Table 12: Evaluation of GUARD's defense against multiple MIAs using GPT-Neo 1.3B model. Performance is measured using TPR@1%FPR scores, where lower values(↓) indicate stronger defense.

| MIAs | PileCC | | | Wiki | | | HackerNews | | | PubMed | | | Arxiv | | | Github | | |
|---|---|---|---|---|---|---|---|---|---|---|---|---|---|---|---|---|---|---|
| | PT | FT | Our | PT | FT | Our | PT | FT | Our | PT | FT | Our | PT | FT | Our | PT | FT | Our |
| Zlib | 0.008 | 0.265 | 0.008 | 0.009 | 0.267 | 0.009 | 0.008 | 0.272 | 0.008 | 0.010 | 0.267 | 0.008 | 0.008 | 0.270 | 0.010 | 0.008 | 0.269 | 0.008 |
| Loss | 0.011 | 0.154 | 0.011 | 0.011 | 0.156 | 0.011 | 0.010 | 0.161 | 0.010 | 0.011 | 0.155 | 0.011 | 0.011 | 0.157 | 0.011 | 0.010 | 0.160 | 0.011 |
| Lowercase | 0.009 | 0.236 | 0.009 | 0.010 | 0.242 | 0.011 | 0.008 | 0.244 | 0.009 | 0.009 | 0.243 | 0.010 | 0.011 | 0.240 | 0.010 | 0.009 | 0.253 | 0.009 |
| Mink | 0.011 | 0.265 | 0.011 | 0.011 | 0.277 | 0.011 | 0.012 | 0.266 | 0.011 | 0.010 | 0.287 | 0.011 | 0.009 | 0.265 | 0.011 | 0.015 | 0.273 | 0.015 |
| Mink++ | 0.014 | 0.286 | 0.014 | 0.015 | 0.280 | 0.014 | 0.014 | 0.279 | 0.014 | 0.012 | 0.243 | 0.012 | 0.016 | 0.252 | 0.016 | 0.021 | 0.268 | 0.021 |
| ReCall | 0.006 | 0.226 | 0.006 | 0.008 | 0.215 | 0.008 | 0.016 | 0.213 | 0.016 | 0.004 | 0.226 | 0.004 | 0.014 | 0.225 | 0.014 | 0.015 | 0.230 | 0.015 |
| Con-ReCall | 0.015 | 0.146 | 0.015 | 0.011 | 0.155 | 0.011 | 0.014 | 0.154 | 0.014 | 0.011 | 0.157 | 0.011 | 0.013 | 0.168 | 0.013 | 0.018 | 0.166 | 0.018 |
| Ratio | 0.005 | 0.754 | 0.005 | 0.006 | 0.728 | 0.006 | 0.017 | 0.669 | 0.017 | 0.015 | 0.743 | 0.015 | 0.007 | 0.742 | 0.007 | 0.018 | 0.756 | 0.017 |
| Self-prompt | 0.012 | 0.772 | 0.012 | 0.007 | 0.783 | 0.007 | 0.008 | 0.759 | 0.009 | 0.006 | 0.774 | 0.006 | 0.005 | 0.770 | 0.005 | 0.019 | 0.769 | 0.018 |

## A.9 SUPPLEMENTARY EXPERIMENTAL RESULTS

### A.9.1 DEFENSE AGAINST EXTRACTION ATTACK

**Connection between extraction and MIA.** A standard extraction pipeline (i) queries the model with generic or style-matched prompts, (ii) collects generated passages, and (iii) runs a membership-inference test to decide whether each passage likely originated from the model's fine-tuning set. Formally, for a generated text $\hat{y} \sim \pi(\cdot \mid x)$, the attacker applies an MIA oracle $\mathcal{M}(\hat{y}) \in \{0, 1\}$ (or a score $\mathcal{M}(\hat{y}) \in [0, 1]$) to filter candidates that appear "in-training." Under this threat model, *reducing MIA accuracy on the fine-tuning distribution directly weakens extraction*, because the attacker's precision/recall in surfacing training texts collapses when $\mathcal{M}$ can no longer distinguish members from non-members.

**Implication for our defense.** Since our method suppresses the membership signal on the fine-tuning data (lower MIA AUC), the attacker's post-generation filter becomes unreliable, which in turn *substantially mitigates extraction* of the fine-tuning corpus.

**Why SOFT fails.** In contrast, models trained with SOFT tend to retain elevated probabilities for training snippets, making them more likely to regenerate near-verbatim passages. Those generations then trigger high MIA scores, enabling the attacker's filter. Consequently, SOFT *does not defend against extraction*: it both facilitates memorized text generation and leaves a strong membership footprint that $\mathcal{M}$ can exploit.

### A.9.2 TPR@1%FPR RESULTS FOR GPT-NEO 1.3B AND QWEN-INSTRUCT 3B MODEL

Tables 12 and 13 report the performance of our proposed **GUARD** defense against 9 different membership inference attacks, evaluated on two model families: GPT-Neo 1.3B and Qwen-Instruct 3B. We measure the attack success rate using **TPR@1%FPR**, where lower values indicate stronger privacy protection.

Across all six datasets (PileCC, Wiki, HackerNews, PubMed, Arxiv, and Github) and all MIA variants, our method consistently reduces the TPR@1%FPR scores to values **close to 0.01**, which approaches the theoretical limit of random guessing. This demonstrates that GUARD is highly effective in mitigating privacy leakage and provides robust generalization across different model architectures and data domains.

### A.9.3 MIA DEFENSE RESULTS OF SMALLER MODEL

Table 14 and Table 15 present the evaluation of GUARD's defense against multiple Membership Inference Attacks (MIAs) across six datasets (PileCC, Wiki, HackerNews, PubMed, Arxiv, GitHub) using GPT-Neo 125M and Qwen-Instruct 1B, respectively.

Table 13: Evaluation of GUARD's defense against multiple MIAs using Qwen-Instruct 3B model. Performance is measured using TPR@1%FPR scores, where lower values(↓) indicate stronger defense.

| MIAs | PileCC | | | Wiki | | | HackerNews | | | PubMed | | | Arxiv | | | Github | | |
|---|---|---|---|---|---|---|---|---|---|---|---|---|---|---|---|---|---|---|
| | PT | FT | Our | PT | FT | Our | PT | FT | Our | PT | FT | Our | PT | FT | Our | PT | FT | Our |
| Zlib | 0.008 | 0.266 | 0.008 | 0.008 | 0.259 | 0.008 | 0.008 | 0.277 | 0.008 | 0.008 | 0.272 | 0.008 | 0.008 | 0.273 | 0.008 | 0.010 | 0.274 | 0.010 |
| Loss | 0.011 | 0.146 | 0.011 | 0.009 | 0.145 | 0.009 | 0.007 | 0.144 | 0.007 | 0.012 | 0.151 | 0.012 | 0.006 | 0.146 | 0.006 | 0.014 | 0.145 | 0.014 |
| Lowercase | 0.009 | 0.225 | 0.009 | 0.005 | 0.224 | 0.005 | 0.005 | 0.228 | 0.005 | 0.007 | 0.226 | 0.007 | 0.013 | 0.225 | 0.013 | 0.015 | 0.224 | 0.015 |
| Mink | 0.011 | 0.278 | 0.011 | 0.010 | 0.279 | 0.011 | 0.008 | 0.285 | 0.008 | 0.012 | 0.281 | 0.012 | 0.014 | 0.283 | 0.014 | 0.021 | 0.282 | 0.021 |
| Mink++ | 0.011 | 0.159 | 0.011 | 0.009 | 0.164 | 0.009 | 0.012 | 0.162 | 0.012 | 0.014 | 0.162 | 0.014 | 0.011 | 0.158 | 0.011 | 0.012 | 0.165 | 0.012 |
| ReCall | 0.006 | 0.164 | 0.006 | 0.005 | 0.165 | 0.005 | 0.007 | 0.163 | 0.007 | 0.008 | 0.166 | 0.008 | 0.008 | 0.164 | 0.008 | 0.032 | 0.165 | 0.032 |
| Con-ReCall | 0.006 | 0.155 | 0.006 | 0.009 | 0.154 | 0.009 | 0.005 | 0.157 | 0.005 | 0.007 | 0.156 | 0.007 | 0.005 | 0.155 | 0.005 | 0.031 | 0.154 | 0.031 |
| Ratio | 0.005 | 0.820 | 0.005 | 0.007 | 0.839 | 0.007 | 0.004 | 0.818 | 0.004 | 0.005 | 0.881 | 0.005 | 0.006 | 0.820 | 0.006 | 0.026 | 0.822 | 0.026 |
| Self-prompt | 0.006 | 0.886 | 0.006 | 0.005 | 0.896 | 0.005 | 0.008 | 0.710 | 0.008 | 0.010 | 0.876 | 0.010 | 0.014 | 0.891 | 0.014 | 0.016 | 0.884 | 0.016 |

Table 14: Evaluation of GUARD's defense against multiple MIAs using GPT-Neo 125m model. Performance is measured using AUC-ROC scores, where lower values(↓) indicate stronger defense.

| MIAs | PileCC | | | Wiki | | | HackerNews | | | PubMed | | | Arxiv | | | Github | | |
|---|---|---|---|---|---|---|---|---|---|---|---|---|---|---|---|---|---|---|
| | PT | FT | Our | PT | FT | Our | PT | FT | Our | PT | FT | Our | PT | FT | Our | PT | FT | Our |
| Zlib | 0.485 | 0.552 | 0.485 | 0.485 | 0.556 | 0.485 | 0.484 | 0.554 | 0.485 | 0.485 | 0.556 | 0.485 | 0.485 | 0.554 | 0.485 | 0.485 | 0.562 | 0.485 |
| Loss | 0.495 | 0.889 | 0.495 | 0.497 | 0.891 | 0.497 | 0.498 | 0.887 | 0.498 | 0.497 | 0.889 | 0.497 | 0.495 | 0.892 | 0.495 | 0.496 | 0.886 | 0.496 |
| Lowercase | 0.494 | 0.896 | 0.494 | 0.495 | 0.895 | 0.495 | 0.496 | 0.882 | 0.495 | 0.495 | 0.894 | 0.495 | 0.494 | 0.887 | 0.494 | 0.501 | 0.901 | 0.502 |
| Mink | 0.493 | 0.928 | 0.495 | 0.496 | 0.916 | 0.495 | 0.495 | 0.929 | 0.495 | 0.494 | 0.933 | 0.494 | 0.496 | 0.930 | 0.496 | 0.495 | 0.927 | 0.496 |
| Mink++ | 0.524 | 0.955 | 0.521 | 0.522 | 0.988 | 0.522 | 0.508 | 0.984 | 0.508 | 0.516 | 0.944 | 0.516 | 0.524 | 0.982 | 0.524 | 0.522 | 0.989 | 0.520 |
| ReCall | 0.505 | 0.968 | 0.505 | 0.502 | 0.991 | 0.502 | 0.504 | 0.996 | 0.504 | 0.507 | 0.994 | 0.503 | 0.499 | 0.995 | 0.499 | 0.499 | 0.995 | 0.499 |
| Con-ReCall | 0.506 | 0.993 | 0.506 | 0.505 | 0.988 | 0.505 | 0.504 | 0.985 | 0.502 | 0.506 | 0.987 | 0.505 | 0.504 | 0.992 | 0.504 | 0.503 | 0.994 | 0.504 |
| Ratio | 0.501 | 0.992 | 0.508 | 0.503 | 0.990 | 0.517 | 0.499 | 0.990 | 0.514 | 0.502 | 0.991 | 0.516 | 0.492 | 0.991 | 0.495 | 0.498 | 0.992 | 0.512 |
| Self-prompt | 0.502 | 0.996 | 0.510 | 0.501 | 0.995 | 0.514 | 0.498 | 0.995 | 0.515 | 0.495 | 0.997 | 0.512 | 0.494 | 0.994 | 0.495 | 0.502 | 0.996 | 0.516 |

Performance is measured using AUC-ROC scores, where lower values indicate stronger defenses. Each table compares results from the pre-trained model (PT), the fine-tuned model (FT), and our GUARD approach (Our).

Across both models, GUARD consistently reduces the AUC-ROC scores compared to FT, demonstrating enhanced robustness to MIAs across all attack types and datasets. Notably, even on larger and more challenging datasets (e.g., PubMed, Arxiv), GUARD maintains strong defensive performance.

### A.9.4 MACHINE UNLEARNING FOR MIAS DEFENSE

Figure 6 illustrates the trade-off between forget quality and model utility using the unlearning setup from prior work. We fine-tune the Qwen-Instruct 3B model on 10k samples from the PileCC dataset, and then perform unlearning on either 100 or 400 samples. The left subfigure shows results for 100 samples, while the right shows results for 400 samples. As the forget quality increases, model utility generally decreases. Notably, completely forgetting 400 samples leads to a near collapse in model performance, highlighting the difficulty of achieving high-quality unlearning without sacrificing utility.

### A.9.5 DIFFERENTIAL PRIVACY FOR MIAS DEFENSE

Table 16 presents the results of applying DP-LoRA for MIA defense under varying noise scales $\epsilon$. Differential privacy (DP) introduces noise to training updates, which mitigates overfitting and thereby reduces susceptibility to membership inference attacks. As shown in the table, smaller values of $\epsilon$ (i.e., stronger privacy) generally correspond to improved defense performance, as indicated by lower AUC-ROC scores across multiple attack methods. However, this improvement comes at

Table 15: Evaluation of GUARD's defense against multiple MIAs using Qwen-Instruct 1B model. Performance is measured using AUC-ROC scores, where lower values(↓) indicate stronger defense.

| MIAs | PileCC | | | Wiki | | | HackerNews | | | PubMed | | | Arxiv | | | Github | | |
|---|---|---|---|---|---|---|---|---|---|---|---|---|---|---|---|---|---|---|
| | PT | FT | Our | PT | FT | Our | PT | FT | Our | PT | FT | Our | PT | FT | Our | PT | FT | Our |
| Zlib | 0.485 | 0.940 | 0.485 | 0.485 | 0.936 | 0.485 | 0.484 | 0.940 | 0.485 | 0.485 | 0.941 | 0.485 | 0.485 | 0.944 | 0.485 | 0.485 | 0.939 | 0.485 |
| Loss | 0.497 | 0.999 | 0.497 | 0.497 | 1.000 | 0.497 | 0.497 | 0.999 | 0.497 | 0.498 | 0.998 | 0.498 | 0.496 | 0.999 | 0.496 | 0.497 | 0.999 | 0.497 |
| Lowercase | 0.495 | 0.954 | 0.495 | 0.496 | 0.967 | 0.496 | 0.494 | 0.950 | 0.494 | 0.495 | 0.949 | 0.495 | 0.496 | 0.953 | 0.496 | 0.502 | 0.965 | 0.502 |
| Mink | 0.496 | 0.999 | 0.496 | 0.497 | 0.996 | 0.497 | 0.496 | 0.999 | 0.496 | 0.498 | 0.998 | 0.498 | 0.498 | 0.998 | 0.498 | 0.497 | 0.997 | 0.497 |
| Mink++ | 0.499 | 0.969 | 0.499 | 0.501 | 0.966 | 0.500 | 0.500 | 0.964 | 0.499 | 0.498 | 0.963 | 0.498 | 0.499 | 0.966 | 0.499 | 0.498 | 0.959 | 0.498 |
| ReCall | 0.498 | 0.977 | 0.498 | 0.499 | 0.979 | 0.499 | 0.498 | 0.976 | 0.498 | 0.502 | 0.976 | 0.502 | 0.497 | 0.976 | 0.497 | 0.496 | 0.972 | 0.498 |
| Con-ReCall | 0.496 | 0.984 | 0.496 | 0.497 | 0.988 | 0.497 | 0.496 | 0.985 | 0.497 | 0.499 | 0.979 | 0.499 | 0.497 | 0.984 | 0.497 | 0.497 | 0.985 | 0.497 |
| Ratio | 0.502 | 0.994 | 0.502 | 0.503 | 0.997 | 0.517 | 0.499 | 0.994 | 0.514 | 0.502 | 0.996 | 0.516 | 0.492 | 0.997 | 0.515 | 0.514 | 0.993 | 0.514 |
| Self-prompt | 0.505 | 0.995 | 0.512 | 0.498 | 0.994 | 0.512 | 0.498 | 0.992 | 0.514 | 0.495 | 0.996 | 0.513 | 0.496 | 0.990 | 0.513 | 0.498 | 0.979 | 0.514 |

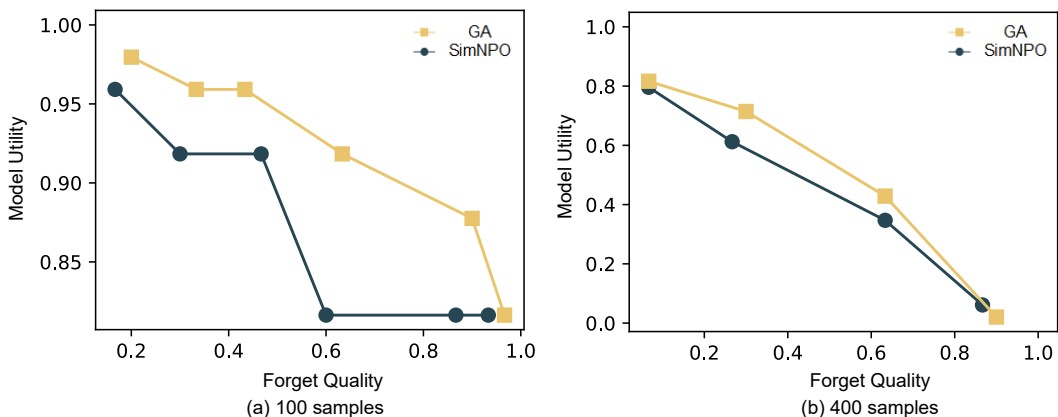

Figure 6: Trade-off between forget quality and model utility for unlearning on the Llama-Instruct 3B model on PileCC.

Table 16: DP-LoRA across different noise scales. Defense effectiveness is evaluated by AUC-ROC against MIAs, where lower values($\downarrow$) indicate stronger defense. Model utility is measured by perplexity, the lower($\downarrow$) the better.

| Methods | $\epsilon$ | Loss | Zlib | Lowercase | Mink | Mink++ | ReCall | CON-ReCall | Ratio | Perplexity |
|---|---|---|---|---|---|---|---|---|---|---|
| pre-trained | | 0.499 | 0.486 | 0.471 | 0.495 | 0.498 | 0.501 | 0.499 | 0.505 | 13.45 |
| | 0.01 | 0.501 | 0.508 | 0.502 | 0.505 | 0.511 | 0.503 | 0.506 | 0.499 | 13.26 |
| | 1 | 0.516 | 0.511 | 0.506 | 0.508 | 0.515 | 0.506 | 0.512 | 0.502 | 13.03 |
| DP-LoRA | 10 | 0.552 | 0.568 | 0.523 | 0.515 | 0.532 | 0.522 | 0.524 | 0.521 | 12.98 |
| | 20 | 0.602 | 0.589 | 0.536 | 0.541 | 0.559 | 0.579 | 0.561 | 0.551 | 12.67 |
| | 60 | 0.625 | 0.616 | 0.605 | 0.556 | 0.607 | 0.633 | 0.572 | 0.563 | 12.55 |
| | 100 | 0.667 | 0.657 | 0.648 | 0.564 | 0.618 | 0.701 | 0.602 | 0.587 | 12.47 |

the cost of model utility, measured by perplexity. For instance, as $\epsilon$ increases from 0.01 to 100, the average AUC-ROC scores degrade significantly, and perplexity drops from 13.26 to 12.47. Notably, when $\epsilon$ is set too high, the privacy guarantee weakens, and defense effectiveness deteriorates. These results highlight the trade-off between privacy and utility when applying DP-based defenses.

### A.9.6 MODEL UTILITY EVALUATION EXAMPLES

We select the `PileCC` and `Wikipedia` datasets for evaluating model utility, as these datasets are relatively general-purpose and not overly complex. In contrast, other datasets such as `PubMed` and `ArXiv` tend to contain longer or more technical content, making them less suitable for our evaluation setting.

To ensure compatibility with the model's input length limitations, we filter the evaluation examples to include only those with a sequence length of fewer than 2048 tokens, aligning with the model's maximum context window.

Fig. 7 presents the original passage along with the generated questions and corresponding gold answers. Fig. 8 provides the detailed evaluation by ChatGPT-4o of the responses produced by the GUARD-enhanced Qwen-Instruct 3B model. Fig. 9 displays the evaluation of the standard fine-tuned Qwen-Instruct 3B model, while Fig. 10 shows the evaluation results for the pre-trained Qwen-Instruct 3B model.

**Passage:** Based out of Los Angeles, we specialize in service and repair of all major home and commercial appliances, A/C and Heating units, including most brands and models. Serving the Greater Los Angeles and San Fernando Valley, see our Service Areas. Our technicians are well experienced and have many years of field work behind them. We offer same day service on most orders. There is no extra charge for evenings, weekends or holidays. We are always in your area, so there is no travel charge! Lastly, we only install brand new, factory recommended parts.

**Question 1:** What is the company's policy on extra charges for evenings, weekends, or holidays?
**Answer:** There is no extra charge for evenings, weekends or holidays.

**Question 2:** What type of parts does the company use for installations?
**Answer:** brand new, factory recommended parts.

Figure 7: Sample passage of PileCC dataset with corresponding question–answer pairs used for response generation.

Evaluation of GUARD Model Responses

**Question 1:** What is the company's policy on extra charges for evenings, weekends, or holidays?
**Gold Answer:** There is no extra charge for evenings, weekends or holidays.
**GUARD Answer:** No extra charges for these times.
**Evaluation:**

- **Helpfulness:** The predicted answer is helpful, as it conveys the correct policy and is a reasonably concise paraphrase of the original statement.
- **Relevance:** The response is relevant and addresses the specific time periods, even though it uses a more general phrase ("these times").
- **Accuracy:** The answer is accurate, though slightly less specific than the gold answer. It omits naming the times but remains faithful in meaning.

**Rating:** [[8]]

**Question 2:** What type of parts does the company use for installations?
**Gold Answer:** brand new, factory recommended parts
**GUARD Answer:** High Quality Parts
**Evaluation:**

- **Helpfulness:** Minimally helpful. The answer is vague and lacks the detailed qualifiers present in the gold answer.
- **Relevance:** Broadly relevant to the topic but fails to reflect the company's stated policy accurately.
- **Accuracy:** Not accurate. It omits key information such as "brand new" and "factory recommended," which are important details.

**Rating:** [[4]]

Figure 8: Example on Fig 7 evaluation of GUARD Qwen 3B model answers on PileCC dataset using GPT-4o scoring.

> ### Evaluation of Fine-Tuned Model Responses
>
> **Question 1:** What is the company's policy on extra charges for evenings, weekends, or holidays?
> **Gold Answer:** There is no extra charge for evenings, weekends or holidays.
> **Fine-tuned Model Answer:** No extra charge is made for evenings, weekends, or holidays. The company policy...
> **Evaluation:**
>
> - **Helpfulness:** Very helpful. The response directly matches the gold answer in meaning and closely mirrors the wording. However, the sentence is incomplete, slightly reducing clarity and polish.
> - **Relevance:** Highly relevant. It precisely addresses the elements asked about in the question and affirms there is no extra charge.
> - **Accuracy:** Factually correct and fully aligned with the passage. The only issue is the unfinished sentence, which affects fluency but not content correctness.
>
> **Rating:** `[[9]]`
>
> **Question 2:** What type of parts does the company use for installations?
> **Gold Answer:** brand new, factory recommended parts
> **Fine-tuned Model Answer:** High quality metal fastener components.
> **Evaluation:**
>
> - **Helpfulness:** Not helpful. The answer introduces specifics not supported by the passage and fails to reflect the key attributes from the gold answer.
> - **Relevance:** Poorly relevant. It answers a different question and does not capture the general policy stated in the original text.
> - **Accuracy:** Inaccurate. The prediction fabricates details (metal fasteners) and omits crucial descriptors like "brand new" and "factory recommended."
>
> **Rating:** `[[2]]`

Figure 9: Example on Fig 7 evaluation of fine-tuned Qwen 3B model answers using GPT-4o scoring.

**Evaluation of pre-trained Model Responses**

**Question 1:** What is the company's policy on extra charges for evenings, weekends, or holidays?
**Gold Answer:** There is no extra charge for evenings, weekends or holidays.
**pre-trained Model Answer:** Evening weekend holiday surcharge policy
**Evaluation:**

- **Accuracy:** The answer is not factual—it reads like a keyword phrase or heading, not a complete or meaningful response.

- **Relevance:** While the phrase includes relevant terms, it fails to express the actual policy being asked for.

- **Helpfulness:** Unhelpful. It does not inform the user whether a surcharge exists or not.

This response seems more like an extracted topic label than a generated answer.
**Rating: [[1]]**

**Question 2:** What type of parts does the company use for installations?
**Gold Answer:** brand new, factory recommended parts
**pre-trained Model Answer:** Manufacturing components
**Evaluation:**

- **Accuracy:** Inaccurate. The term "manufacturing components" is vague and does not reflect the qualities described in the passage.

- **Relevance:** Only loosely relevant. It misses the focus of the question on part specifications.

- **Helpfulness:** Not helpful. It fails to provide the essential details (brand new, factory recommended) that directly answer the question.

The model offers a generic term instead of extracting or paraphrasing the precise answer.
**Rating: [[1]]**

Figure 10: Example on Fig 7 evaluation of pre-trained Qwen 3B model responses using GPT-4o scoring.

