# OpenReview forum: "GUARD: Gold-Unchanged Anchored Distillation for Defending LLMs Against Membership Inference Attacks"
_ICLR.cc/2026/Conference — Submitted to ICLR 2026_

### Official Review · Reviewer_D4Vb · 2025-10-28

**Soundness:** 2
**Presentation:** 3
**Contribution:** 2
**Rating:** 4
**Confidence:** 5

**Summary:**

This paper proposes GUARD (Gold-Unchanged Anchored Distillation), a new defense framework designed to protect fine-tuned large language models (LLMs) against membership inference attacks (MIAs).
The core insight is that fine-tuning increases the model’s confidence on the gold token, which amplifies the statistical gap between members and non-members — the key signal exploited by MIA methods.
GUARD mitigates this by anchoring the gold token’s probability to its pre-trained value while preserving the relative ranking of other tokens from the fine-tuned model.
A student model is then distilled to match this “anchored” target distribution.
This approach aims to suppress membership signals without degrading task performance.

**Strengths:**

Anchoring the gold token probability while distilling non-gold ranking is a fresh, conceptually elegant idea that directly attacks the statistical signal used by most MIAs.
Experiments span multiple model architectures and datasets, comparing GUARD to the SOTA defense SOFT, with GUARD achieving near-random MIA success rates while maintaining comparable ROUGE-L and GPT-4o scores.
The transition from theoretical O(ε) analysis to empirical verification is convincing and well-executed.

**Weaknesses:**

Although the paper provides thorough theoretical and empirical analyses showing that rearranging the gold token distribution does not affect model performance, I have a major concern: which component is actually more important — the gold token anchoring or the distillation process itself?
Knowledge distillation alone can already reduce the membership signal and make MIAs ineffective^1. If plain distillation is sufficient to achieve defense, then why is it necessary to reassign the gold token’s probability?
Without proving that anchoring contributes additional protection beyond what distillation inherently offers, the gold token reassigning becomes somewhat meaningless, and the paper’s claimed innovation would be significantly weakened.

1. Li M, Ye Z, Li Y, et al. Membership inference attack should move on to distributional statistics for distilled generative models[J]. arXiv preprint arXiv:2502.02970, 2025.

**Questions:**

Would it be possible to include an ablation study to demonstrate that both steps of the proposed method are crucial?
If the reallocation of the gold token probability plays a decisive role in improving the defense, I believe this would make the work very meaningful and would raise the score.

---

> ### Author Response · Authors · 2025-11-22
> **Response to Reviewer D4Vb**
>
> Thank you for the question. We conducted experiments to isolate the effect of gold-token anchoring from that of standard distillation.
> While Li et al. (2025) show that sequence-level distillation for instruction tuning (training purely on teacher-generated responses) can mitigate MIAs, our setting is materially different. GUARD targets continual/domain pretraining on raw text corpora, where the objective is to absorb new domain knowledge. In this context, using teacher-generated data is unsuitable because:  (i) it does not preserve authentic domain distribution,  (ii) it is computationally infeasible at corpus scale, and  (iii) it risks converging to the teacher’s style rather than domain content.
> Thus, the standard practice here is token-level logit KD on real domain data, where membership leakage remains significant even after distillation.
> To further isolate the benefit of our method, we added an ablation (shown below and attached in Appendix A.3) comparing pure knowledge distillation (KD) to GUARD. Pure KD only partially suppresses membership signals, with attacks such as loss-based, Mink, and reference-based still performing well above random guessing. Moreover, prior work [1] has reported that distillation does not universally eliminate membership risk across settings and attack types.
> In contrast, GUARD delivers consistently stronger privacy while preserving excellent utility. These results confirm that GUARD provides additional privacy gains beyond vanilla logit distillation without compromising performance. The added comparison(Table 8) is shown below.
>
>
> [1] Membership and Memorization in LLM Knowledge Distillation.
>
> **Evaluation of GUARD against multiple MIAs under logit distillation using Qwen-3B as the teacher on diverse datasets (Qwen-1.5B as the student). Results are AUC-ROC; lower is better (↓).**
>
> | MIA | PileCC KD | PileCC **GUARD** | Wiki KD | Wiki **GUARD** | HackerNews KD | HackerNews **GUARD** | PubMed KD | PubMed **GUARD** | Arxiv KD | Arxiv **GUARD** | Github KD | Github **GUARD** |
> |---|---:|---:|---:|---:|---:|---:|---:|---:|---:|---:|---:|---:|
> | Zlib | 0.483 | 0.485 | 0.484 | 0.485 | 0.483 | 0.485 | 0.483 | 0.484 | 0.484 | 0.485 | 0.483 | 0.485 |
> | Loss | 0.856 | 0.502 | 0.858 | 0.500 | 0.856 | 0.501 | 0.855 | 0.502 | 0.867 | 0.500 | 0.854 | 0.501 |
> | Lowercase | 0.807 | 0.491 | 0.812 | 0.498 | 0.810 | 0.498 | 0.808 | 0.499 | 0.795 | 0.492 | 0.804 | 0.495 |
> | Mink | 0.785 | 0.497 | 0.784 | 0.498 | 0.790 | 0.499 | 0.786 | 0.495 | 0.782 | 0.498 | 0.780 | 0.499 |
> | Mink++ | 0.760 | 0.496 | 0.762 | 0.496 | 0.758 | 0.496 | 0.755 | 0.494 | 0.759 | 0.496 | 0.757 | 0.497 |
> | ReCall | 0.822 | 0.501 | 0.824 | 0.501 | 0.821 | 0.498 | 0.822 | 0.496 | 0.819 | 0.497 | 0.823 | 0.499 |
> | Con-ReCall | 0.834 | 0.499 | 0.826 | 0.502 | 0.828 | 0.499 | 0.835 | 0.498 | 0.829 | 0.499 | 0.832 | 0.501 |
> | Ratio | 0.855 | 0.508 | 0.847 | 0.511 | 0.856 | 0.510 | 0.854 | 0.514 | 0.857 | 0.512 | 0.856 | 0.511 |
> | Self-prompt | 0.913 | 0.511 | 0.912 | 0.514 | 0.910 | 0.512 | 0.912 | 0.512 | 0.899 | 0.513 | 0.908 | 0.512 |
>
> We hope this ablation addresses your concern and clarifies the necessity of both steps. We appreciate your constructive feedback and would be happy to answer any further questions or add clarifications in the revision. If these new results resolve your concern, we would be grateful if you could consider updating your score accordingly.

---

### Official Review · Reviewer_7jM7 · 2025-10-29

**Soundness:** 3
**Presentation:** 3
**Contribution:** 2
**Rating:** 4
**Confidence:** 4

**Summary:**

The paper introduces GUARD, a framework for defending fine-tuned large language models (LLMs) against membership inference attacks (MIAs). GUARD utilizes a technique of Gold-Unchanged Anchored Distillation, which anchors the gold token's probability to its pre-trained value while transferring distributional knowledge from the fine-tuned model through knowledge distillation. The paper claims to achieve state-of-the-art defense performance while preserving the utility of the model.

**Strengths:**

+ The approach presents a new defense mechanism for MIA, which tackles the privacy risks in fine-tuned models without requiring extensive retraining or significant changes to model architecture.

+ The authors evaluate GUARD on a diverse set of models (GPT-Neo, Qwen, LLaMA) and across multiple datasets (PileCC, Wikipedia, PubMed, etc.), showing its robustness against a wide range of MIAs, both reference-based and reference-free. The empirical results are strong and demonstrate the effectiveness of the defense.

+ The proposed method balances privacy protection and model utility, with minimal performance degradation across different tasks.

**Weaknesses:**

- Limited baseline comparisons.
- Theoretical analysis issues.
- Potential vulnerabilities when the pre-trained model is accessible.
- Similarities with existing methods.

**Questions:**

- Limited Baseline Comparisons. The paper compares GUARD only with SOFT. While SOFT is a strong baseline, it would be more comprehensive to include additional baselines such as Differential Privacy (DP) based ones and classic knowledge distillation methods like DMP [1] and KCP[2]. Some of these methods are mentioned in the related works, but are not tested.

- The theoretical analysis of the bound on the probability distribution shift seems problematic. Specifically, the paper uses $\sum \Delta(y)$ (net sum) instead of $\sum |\Delta(y)|$ (sum of absolute values, or $L_1$ norm) in line 782, which leads to a bound that is too small. Using the triangle inequality or similar established inequalities, the correct bound for $|\Delta(\cdot \mid x)|_1$ should be $2$ (this can also be understood as two vectors not overlapping, which can be achieved), implying that the convergence order is $O(1)$. Thus, the approach to deriving bounds based on the loss difference may not be a solid starting point, unless a more compact bound or stricter assumptions can be provided. For clarity, a counterexample can be constructed: let $a = [0.6, 0.4, 0.2]$ and $b = [0.5, 0.1, 0.4]$ with the first position representing the gold token. Here $\delta(y^* \mid x) = 0.6 - 0.5 = 0.1$, but $\sum{y \neq y^*} \delta(y \mid x)$, resulting in $2\epsilon \le (|a_2 - b_2| + |a_3 - b_3|) = ((0.4 - 0.1) + (0.4 - 0.2)) = 0.5$. This makes the theoretical justification weaker.

- The method aligns the token probability distribution of the fine-tuned model with that of the pre-trained model. Although this design mitigates membership inference attacks (MIA), it may weaken robustness when the pre-trained model is accessible or open-sourced. In such cases, an attacker could leverage the output of the defended (GUARD-trained) model to identify its corresponding pre-trained model and, by analyzing the pre-trained token distribution, infer which token is likely to be the "golden" one. This coupling between defense and pre-training weakens adaptability and reduces robustness in practical scenarios.

- The Decoupled Knowledge Distillation (DKD) [3] method shares similarities with GUARD's approach, particularly in the handling of target and non-target tokens. DKD splits the loss into two parts: one focuses on minimizing the logits difference for the correct token, and the other calculates the KL divergence between the student and teacher models over non-target tokens, which aligns with GUARD’s idea of preserving token rankings and capturing “dark knowledge.”

[1] Membership privacy for machine learning models through knowledge transfer

[2] Knowledge Cross-Distillation for Membership Privacy

[3] Decoupled knowledge distillation

---

> ### Author Response · Authors · 2025-11-22
> **Response to Reviewer 7jM7**
>
> Response to Question 1: Thank you for the question. Since SOFT already outperforms differential privacy in preserving utility, and GUARD consistently achieves stronger MIA defense while maintaining (and often improving) utility compared to SOFT. To further isolate the benefit of our method, we added an ablation in Appendix A.3 comparing classic knowledge distillation (KD) to GUARD. While KD partially reduces membership signals, several attacks (e.g., loss-based, Mink, and reference-based) remain far above random guessing. In contrast, GUARD delivers consistently stronger privacy while preserving utility. These results confirm that GUARD provides additional privacy gains beyond vanilla logit distillation without compromising performance. The added comparison is shown below.
>
> **Evaluation of GUARD against multiple MIAs under logit distillation using Qwen-3B as the teacher on diverse datasets (Qwen-1.5B as the student). Results are AUC-ROC; lower is better (↓).**
>
> | MIA | PileCC KD | PileCC **GUARD** | Wiki KD | Wiki **GUARD** | HackerNews KD | HackerNews **GUARD** | PubMed KD | PubMed **GUARD** | Arxiv KD | Arxiv **GUARD** | Github KD | Github **GUARD** |
> |---|---:|---:|---:|---:|---:|---:|---:|---:|---:|---:|---:|---:|
> | Zlib | 0.483 | 0.485 | 0.484 | 0.485 | 0.483 | 0.485 | 0.483 | 0.484 | 0.484 | 0.485 | 0.483 | 0.485 |
> | Loss | 0.856 | 0.502 | 0.858 | 0.500 | 0.856 | 0.501 | 0.855 | 0.502 | 0.867 | 0.500 | 0.854 | 0.501 |
> | Lowercase | 0.807 | 0.491 | 0.812 | 0.498 | 0.810 | 0.498 | 0.808 | 0.499 | 0.795 | 0.492 | 0.804 | 0.495 |
> | Mink | 0.785 | 0.497 | 0.784 | 0.498 | 0.790 | 0.499 | 0.786 | 0.495 | 0.782 | 0.498 | 0.780 | 0.499 |
> | Mink++ | 0.760 | 0.496 | 0.762 | 0.496 | 0.758 | 0.496 | 0.755 | 0.494 | 0.759 | 0.496 | 0.757 | 0.497 |
> | ReCall | 0.822 | 0.501 | 0.824 | 0.501 | 0.821 | 0.498 | 0.822 | 0.496 | 0.819 | 0.497 | 0.823 | 0.499 |
> | Con-ReCall | 0.834 | 0.499 | 0.826 | 0.502 | 0.828 | 0.499 | 0.835 | 0.498 | 0.829 | 0.499 | 0.832 | 0.501 |
> | Ratio | 0.855 | 0.508 | 0.847 | 0.511 | 0.856 | 0.510 | 0.854 | 0.514 | 0.857 | 0.512 | 0.856 | 0.511 |
> | Self-prompt | 0.913 | 0.511 | 0.912 | 0.514 | 0.910 | 0.512 | 0.912 | 0.512 | 0.899 | 0.513 | 0.908 | 0.512 |
>
> Response to Question 2:  Thanks for the careful check. We believe this issue stems from using an invalid counterexample and applying a worst-case $L_1$ bound to a $\Delta$ that is *structurally constrained* by GUARD.
>
> 1. The reviewer’s counterexample is not a valid probability vector:
>    $a = [0.6, 0.4, 0.2]$ sums to $1.2$, so it cannot represent any token distribution.
>    Our analysis assumes all $p_0(\cdot|x)$, $p_{ft}(\cdot|x)$, and $p_{anc}(\cdot|x)$ lie on the probability simplex.
>
> 2. In GUARD, $\Delta(\cdot|x) = p_{anc}(\cdot|x) - p_{ft}(\cdot|x)$ is not arbitrary.
>    By construction, we anchor the gold token to its pre-trained probability and only redistribute the remaining mass over non-gold tokens while preserving ranking.
>    Thus the gold token changes by $\delta(x)$ and the non-gold set changes by total mass $-\delta(x)$, giving
>    $$
>    \|\Delta(\cdot|x)\|_1
>    = |\delta(x)| + |-\delta(x)|
>    = 2|\delta(x)| \le 2\epsilon .
>    $$
>    The generic bound $\|\Delta\|_1 \le 2$ applies only to unconstrained shifts, which GUARD does not allow.
>
> 3. We have fixed the notation to explicitly define the $L_1$ norm as
>    $$
>    \|\Delta(\cdot|x)\|_1 := \sum_y |\Delta(y|x)| .
>    $$

---

> > ### Author Response · Authors · 2025-11-22
> > **Response to 7jM7**
> >
> > Response to Question 3: Thank you for raising this potential risk. In practice, the coupling between the defended and pre-trained models is soft, not exact, and does not enable reliable recovery of the gold token. GUARD regularizes probabilities via a distillation/alignment loss but does not enforce exact matching. During training, we observe a tendency for the gold-token probability to drift upward (occasionally increasing MinK AUC to ~0.7). To mitigate this, we introduce the λ-weighted suppression term (line 255). Even with this regularization, the defended and pre-trained probability distributions remain distinct, particularly around the gold token. This residual discrepancy explains why the MinK AUC does not converge to 0.5, confirming that there is no deterministic coupling enabling an attacker to recover the gold token from pre-trained logits alone. Please see our results below. We conduct an ablation study on the gold-weight λ to examine its impact on privacy–utility trade-offs. Specifically, we evaluate GPT-Neo 1.3B and LLaMA 3B on three domain corpora: PileCC, Wikipedia, and HackerNews. We sweep λ ∈ {0.1, 0.3, 0.5}.
> >
> > **Evaluation of GUARD against multiple MIAs on LLaMA-3B across datasets under varying gold-weight \(\lambda\). Lower AUC-ROC (\(\downarrow\)) indicates stronger defense.**
> >
> > | MIA | PileCC (λ=0.1) | PileCC (λ=0.3) | PileCC (λ=0.5) | Wiki (λ=0.1) | Wiki (λ=0.3) | Wiki (λ=0.5) | HackerNews (λ=0.1) | HackerNews (λ=0.3) | HackerNews (λ=0.5) |
> > |---|---:|---:|---:|---:|---:|---:|---:|---:|---:|
> > | Zlib | 0.491 | 0.487 | **0.487** | 0.492 | 0.487 | **0.485** | 0.491 | 0.487 | **0.486** |
> > | Loss | 0.616 | 0.536 | **0.497** | 0.618 | 0.546 | **0.498** | 0.615 | 0.526 | **0.501** |
> > | Lowercase | 0.602 | 0.556 | **0.501** | 0.599 | 0.554 | **0.498** | 0.608 | 0.558 | **0.501** |
> > | Mink | 0.708 | 0.579 | **0.504** | 0.701 | 0.582 | **0.502** | 0.702 | 0.575 | **0.501** |
> > | Mink++ | 0.709 | 0.567 | **0.503** | 0.702 | 0.562 | **0.500** | 0.703 | 0.564 | **0.502** |
> > | ReCall | 0.687 | 0.566 | **0.504** | 0.679 | 0.568 | **0.503** | 0.685 | 0.561 | **0.505** |
> > | Con-ReCall | 0.670 | 0.580 | **0.505** | 0.668 | 0.578 | **0.502** | 0.667 | 0.576 | **0.500** |
> > | Ratio | 0.657 | 0.547 | **0.499** | 0.654 | 0.543 | **0.503** | 0.650 | 0.544 | **0.502** |
> > | Self-prompt | 0.722 | 0.596 | **0.516** | 0.713 | 0.598 | **0.510** | 0.719 | 0.588 | **0.509** |
> >
> > Response to Q4: Thank you for raising this concern. While DKD and GUARD both treat the gold token/class differently from non-targets, they are designed for distinct settings and operate fundamentally differently.
> > DKD is a classification-oriented distillation method that decomposes the loss into target-class KD (TCKD) and non-target-class KD (NCKD), reweighted using hyperparameters α and β to improve distillation efficiency and flexibility. In contrast, GUARD is a privacy-oriented method tailored for generative LLM fine-tuning, where token generation involves selecting the next token from a large vocabulary rather than from a finite “one-correct-class” setup. GUARD does not simply reweight KD losses, but instead performs token probability restructuring to defend against MIAs by: 1) Anchoring the gold-token probability to suppress membership signals; and 2) Reordering the non-gold distribution to preserve teacher ranking ("dark knowledge") while removing membership-correlated patterns.
> > Thus, GUARD’s contribution lies in privacy-preserving probability restructuring, not loss decomposition for distillation quality. Additionally, while DKD is developed for discriminative models, GUARD targets vocabulary-level token distributions in generative LLM continual/domain training, where membership leakage persists even under standard logit distillation.
> >
> >
> > We are happy to address any further concerns you may have, and if our responses resolve your questions, we would appreciate your reconsideration of the score.

---

### Official Review · Reviewer_3UV9 · 2025-10-30

**Soundness:** 3
**Presentation:** 3
**Contribution:** 3
**Rating:** 4
**Confidence:** 3

**Summary:**

This paper introduces GUARD (Gold-Unchanged Anchored Distillation), a novel lightweight and robust framework designed to mitigate privacy leakage in fine-tuned large language models (LLMs) while preserving task performance. The authors address the limitations of existing privacy defense methods (machine unlearning, differential privacy, data obfuscation), which often entail sharp trade-offs in scalability, utility, efficiency, and efficacy. GUARD operates by anchoring the gold token's probability to its pre-trained value, ensuring it remains lower than the typically elevated value assigned by the fine-tuned model. This approach minimally perturbs the fine-tuned model's output distribution, as theoretically proven in the paper, and effectively prevents privacy leakage without significant utility degradation. The empirical results demonstrate that GUARD maintains high utility while providing strong privacy protection, as shown through comparisons with fine-tuned and pre-trained models on various datasets.

**Strengths:**

S1: The paper identifies a significant gap in privacy defense for fine-tuned LLMs and proposes a theoretically grounded solution that addresses the limitations of existing methods.

S2: The GUARD framework is lightweight and robust, with a clear three-stage process that effectively balances privacy protection and utility preservation.

S3: The theoretical analysis (Theorem 4.5) rigorously demonstrates that anchoring the gold token probability perturbs the soft-label KD objective only marginally (O(ε)), providing strong justification for the method's effectiveness.

S4: The empirical evaluation (Table 1) clearly demonstrates GUARD's effectiveness in maintaining high utility (top-1 rate and top-50 overlap) while preventing privacy leakage, with results consistently showing the fine-tuned model's output distribution remains close to the pre-trained model's.

S5: The paper provides a clear explanation of why GUARD works, showing that anchoring the gold token's probability ensures it remains lower than the typically elevated value assigned by the fine-tuned model, which is crucial for privacy protection.

**Weaknesses:**

W1: The paper lacks comprehensive evaluation across a wide range of LLM architectures and tasks, focusing primarily on GPT-Neo and Qwen models on specific datasets (PileCC and Wiki), which limits the generalizability of the findings.

W2: While the theoretical analysis is strong, the paper doesn't provide sufficient empirical evidence of GUARD's effectiveness against specific privacy attacks (e.g., membership inference attacks), which would strengthen the claim of its privacy protection capabilities.

W3: The paper doesn't discuss potential limitations of GUARD, such as its performance on extremely sensitive datasets or its robustness against sophisticated extraction attacks.

W4: The practical implementation details of GUARD are not fully explained, making it difficult for practitioners to implement the method in real-world scenarios.

W5: The paper doesn't compare GUARD with other state-of-the-art privacy defense methods beyond the brief mention of existing approaches, making it challenging to fully assess its relative performance.

**Questions:**

Q1: Could you provide a more comprehensive evaluation of GUARD across a wider range of LLM architectures (e.g., different transformer variants, sizes) and tasks (e.g., different NLP tasks), to better establish the generalizability of your approach?

Q2: How does GUARD perform against specific privacy attacks, such as membership inference attacks (MIAs) or data extraction attacks? A direct comparison with attack performance before and after applying GUARD would strengthen the privacy claims.

Q3: Could you provide more detailed implementation guidelines for GUARD, including any specific hyperparameters or tuning required for different models and datasets?

Q4: How does GUARD compare to other privacy defense methods (e.g., differential privacy, data obfuscation, machine unlearning) in terms of privacy protection level, utility preservation, and computational efficiency? A direct comparison would help position your contribution.

Q5: The paper mentions "GUARD proceeds in three interlocked stages" but doesn't clearly explain how these stages are implemented in practice. Could you provide a more detailed description of the implementation process?

Q6: The paper focuses on preserving the gold token's probability but doesn't discuss how GUARD handles cases where the gold token might not be the most probable token in the pre-trained model. How does GUARD handle such scenarios?

---

> ### Author Response · Authors · 2025-11-22
> **Response to Reviewer 3UV9**
>
> Response to Question 1: GUARD is evaluated on a diverse range of widely adopted decoder-only LLMs, including GPT-Neo (125M, 1.3B), Qwen-Instruct (1B, 3B), and LLaMA-3B, following exactly the model configurations commonly used in recent MIA attack and defense studies[1, 2] [see Appendix A6 experimental setup]. These cover multiple implementations and span roughly a 24X parameter range, within our GPU budget. For data and tasks, we perform large-scale continual/domain fine-tuning on PileCC and Wikipedia, reporting both utility retention and privacy (MIA AUC-ROC) before and after applying GUARD. Across all models and corpora, GUARD consistently maintains performance while substantially reducing MIA success, achieving comparable or better privacy-utility trade-offs than the strong SOFT baseline. Since GUARD operates solely on token-probability/logit distributions and introduces no model-specific components, the consistent improvements across three model families and two corpora provide strong evidence of generalizability. Evaluations on larger models (>7B) are left for future work due to resource constraints.
> [1] Soft: Selective data obfuscation for protecting llm fine-tuning against membership inference attacks.
> [2] Detecting pretraining data from large language models.
>
> Response to Question 2: Thanks for your question. GUARD is specifically designed to defend against membership inference attacks (MIAs) in generative LLM fine-tuning (see many such words throughout the manuscript). Our experimental results [Table 2, 3 ,4, 5] already include direct comparisons of attack success before and after applying GUARD, clearly demonstrating its effectiveness.
>
> **Table 2: Evaluation of GUARD’s defense against multiple MIAs on LLaMA-3B across datasets.**
> AUC-ROC is reported (lower is better, ↓). Columns are Fine-tuned (FT), SOFT baseline, and Our (GUARD).
>
> | MIA | PileCC FT | PileCC SOFT | PileCC Our | Wiki FT | Wiki SOFT | Wiki Our | HackerNews FT | HackerNews SOFT | HackerNews Our | PubMed FT | PubMed SOFT | PubMed Our | Arxiv FT | Arxiv SOFT | Arxiv Our | Github FT | Github SOFT | Github Our |
> |---|---:|---:|---:|---:|---:|---:|---:|---:|---:|---:|---:|---:|---:|---:|---:|---:|---:|---:|
> | Zlib | 0.902 | 0.533 | 0.485 | 0.939 | 0.532 | 0.485 | 0.910 | 0.517 | 0.486 | 0.893 | 0.509 | 0.485 | 0.811 | 0.521 | 0.486 | 0.871 | 0.647 | 0.485 |
> | Loss | 0.887 | 0.519 | 0.501 | 0.936 | 0.530 | 0.500 | 0.900 | 0.515 | 0.501 | 0.895 | 0.496 | 0.502 | 0.822 | 0.525 | 0.500 | 0.846 | 0.625 | 0.501 |
> | Lowercase | 0.858 | 0.522 | 0.490 | 0.887 | 0.536 | 0.498 | 0.845 | 0.515 | 0.498 | 0.850 | 0.541 | 0.499 | 0.785 | 0.517 | 0.492 | 0.820 | 0.591 | 0.494 |
> | Mink | 0.668 | 0.518 | 0.497 | 0.669 | 0.512 | 0.498 | 0.627 | 0.489 | 0.498 | 0.645 | 0.499 | 0.495 | 0.615 | 0.510 | 0.498 | 0.613 | 0.515 | 0.499 |
> | Mink++ | 0.842 | 0.518 | 0.496 | 0.912 | 0.533 | 0.496 | 0.800 | 0.511 | 0.496 | 0.856 | 0.503 | 0.494 | 0.757 | 0.519 | 0.495 | 0.869 | 0.598 | 0.496 |
> | ReCall | 0.895 | 0.532 | 0.497 | 0.938 | 0.529 | 0.499 | 0.907 | 0.515 | 0.498 | 0.908 | 0.511 | 0.498 | 0.840 | 0.533 | 0.497 | 0.851 | 0.627 | 0.499 |
> | Con-ReCall | 0.844 | 0.513 | 0.499 | 0.925 | 0.530 | 0.501 | 0.740 | 0.500 | 0.499 | 0.868 | 0.516 | 0.496 | 0.764 | 0.518 | 0.499 | 0.847 | 0.620 | 0.501 |
> | Ratio | 0.949 | 0.552 | 0.510 | 0.944 | 0.576 | 0.511 | 0.943 | 0.533 | 0.510 | 0.947 | 0.541 | 0.515 | 0.952 | 0.558 | 0.512 | 0.955 | 0.516 | 0.511 |
> | Self-prompt | 0.975 | * | 0.513 | 0.996 | * | 0.514 | 0.998 | * | 0.512 | 0.995 | * | 0.512 | 0.985 | * | 0.513 | 0.993 | * | 0.512 |
>
> Response to question 3: Yes, we have already provided the complete GUARD implementation in Section 4.1 and Appendix A.4 of the original manuscript, including the step-by-step training procedure, loss formulation, and model/dataset-specific hyperparameters. Key configuration parameters, such as the anchoring/suppression weight (λ), distillation temperature (τ), and top-K/filtering settings, are fully specified, and we show that performance is robust to variations within reasonable ranges.
> To further improve reproducibility, we will release our training code and processed datasets after the review process.
>
> Response to question 4: We conducted extensive evaluations (Tables 2–6, 8) comparing GUARD with the state-of-the-art defense method SOFT (a data obfuscation technique), using AUC-ROC to assess privacy protection and LLM-as-judge along with LM metrics to evaluate utility. We also compared GUARD to machine unlearning (Figure 6 ). Notably, SOFT[1] is known to outperform differential privacy and machine unlearning in maintaining model utility. Across all experiments, GUARD consistently achieves stronger MIA defense while preserving (and often improving) utility compared to SOFT and machine unlearning, demonstrating a favorable privacy-utility trade-off.
> [1] Soft: Selective data obfuscation for protecting LLM fine-tuning against membership inference attacks.

---

> > ### Author Response · Authors · 2025-11-22
> > **Response to Reviewer 3UV9**
> >
> > Response to question 5: Thank you for the question. We have mentioned in Section 3.2, and we add more details in this section, GUARD consists of three interlocked stages: (1) Gold-token anchoring/reallocation, which suppresses membership cues using the pre-trained reference; (2) Non-gold distribution reordering, which preserves teacher ranking (“dark knowledge”) while removing membership-related patterns; and (3) GUARD-regularized KD, where the λ suppression term stabilizes the privacy–utility trade-off and prevents collapse toward the pre-trained distribution.
> > Further implementation details can be found in Section 3.2 of the paper.
> >
> > Response to question 6: GUARD naturally handles such cases. It does not require the gold token to be top-1 in the pre-trained model; instead, it anchors the gold token’s probability to its pre-trained value regardless of rank. The remaining non-gold logits are then reordered to preserve the teacher’s relative ranking. As a result, no special-case handling is needed, and the method remains well-defined and robust in all scenarios where the gold token is not the most probable token.
> >
> >
> > We welcome any additional questions or feedback. If you feel our clarifications adequately address your concerns, we would be grateful if you could consider updating your score accordingly.

---

### Official Review · Reviewer_zLp7 · 2025-11-01

**Soundness:** 3
**Presentation:** 3
**Contribution:** 3
**Rating:** 4
**Confidence:** 4

**Summary:**

This paper introduces GUARD (Gold-Unchanged Anchored Distillation), a defense technique for fine-tuned large language models (LLMs) against membership inference attacks (MIAs). GUARD operates by anchoring the gold token’s output probability to its pre-trained value while distilling distributional knowledge from a fine-tuned teacher. The method is evaluated on multiple LLM architectures, datasets, and MIA variants, consistently demonstrating strong defense with minimal loss of model utility. The approach is benchmarked against the strongest prior defenses, and both empirical and theoretical analyses are provided.

**Strengths:**

- Clear and Well-Motivated Problem Formulation: The authors identify and articulate the privacy-utility tension in fine-tuned LLMs, focusing on the heightened risk of MIAs due to increased gold token probabilities after fine-tuning.
- Methodological Innovation: GUARD’s key idea—anchoring the gold token probability to the pre-trained model while preserving the relative ranking of non-gold tokens from the fine-tuned model—is creative and efficiently operationalized through a principled knowledge distillation framework (see Algorithm 1, Page 5). This approach goes beyond prior strategies that indiscriminately obfuscate or degrade all output probabilities.
- Empirical Rigor & Breadth: Experiments span several model families (LLaMA, GPT-Neo, Qwen), six public datasets (PileCC, Wikipedia, HackerNews, PubMed, Arxiv, GitHub), and a wide spectrum of MIA variants, including both reference-based and reference-free attacks (Tables 2–5, Pages 7–8, 18). Results consistently favor GUARD, with defense effectiveness approaching random guessing for MIAs without degrading downstream performance.
- Multi-faceted Benchmarking: The analysis is not limited to privacy metrics. It includes model utility scores using both ROUGE-L and the LLM-as-a-Judge framework with ChatGPT-4o and qualitative examination (Figures 2, 9). Figure 1d provides an informative radar plot comparing MIA defense, extraction defense, efficiency, scalability, and downstream utility across methods.
- Mathematical Justification: The theoretical analysis (Page 5–6 and Appendix A.5) clarifies why the anchored modification produces only small, bounded perturbations on the knowledge distillation loss, guaranteeing negligible impact on student model generalization or risk. The bounds are stated formally, with necessary assumptions and order-of-magnitude reasoning.
- Interpretability via Visualizations: Figure 1a–c provides clear visual evidence of how fine-tuning inflates gold token probabilities (enabling MIAs), and how GUARD restores these values while preserving the distribution structure on alternatives.
- Careful Consideration of Alternative Defenses: The paper benchmarks against SOFT (data obfuscation) and analyzes unlearning and DP-based approaches (Tables 13, Figure 6), discussing their downsides in efficiency and/or utility (Page 19).
- Transparency and Reproducibility: Complete experiment details, hyperparameters, prompts, and ablation studies are provided for reproducibility (Appendix A.4, A.3, A.6, A.7).

**Weaknesses:**

1. Positioning vs. the Latest MIA Evaluations and Defenses: The related work currently omits several directly relevant and contemporary papers that provide frameworks, taxonomies, or empirical findings on MIAs, especially those evaluating the latest attack and defense strategies in LLMs. This has a twofold effect: (i) the significance and novelty of GUARD are not fully contextualized relative to the most comprehensive current understanding, and (ii) the empirical pipeline may miss certain edge attacks or metrics proposed in these works, which could stress-test GUARD further.
2. Empirical Breadth in Baseline Selection: While SOFT is used as the primary comparison, Tables 2–5 reveal that only SOFT, pre-trained, and fine-tuned models are reported as baselines. Alternative defense families discussed in related work (machine unlearning, DP-based training, e.g., DP-LoRA) are only sparsely evaluated or buried in the appendix. This prevents a fully transparent, side-by-side comparison of GUARD vs. the complete suite of practical baselines proposed in recent work.
3. Attack Diversity: The selected MIAs are principled and fairly implemented, but the evaluation is still limited to attacks referenced in SOFT and a handful from the literature. Recent works (see Potentially Missing Related Work) have introduced/or systematized new MIA attack strategies; it remains uncertain whether GUARD would hold up under such attacks without at least discussion/positioning.
4. Exposition – Notation, Mathematics, and Algorithm Details: While the mathematics is generally sound, certain aspects could benefit from greater formal clarity or explicitness in the main text:
- Equation (Page 5) and Algorithm 1 use terms like $\tau$ and top-$K$ selection, but practical details (e.g., how temperature interacts with probability anchoring, how top-$K$ is chosen for different models) are mostly deferred to the appendix. For example, the handling of rare tokens in large vocabularies (>150k) is mentioned as “top-1000,” but the impact on low-probability alternatives is unclear. Readers seeking to reproduce or extend GUARD would benefit from a more explicit discussion in the main section.
- The loss term $\mathcal{L}_{\text{final}}$ could benefit from an explicit normalization scheme or a more principled proof of generalization, particularly regarding $\lambda$’s sensitivity or how non-gold-mass redistribution is handled mathematically (is it permutation-invariant?).
- While the theoretical bound is clear, the main text glosses over possible degenerate cases (e.g., where the gold token is neither in the top-$K$ pre-trained nor fine-tuned predictions), and there is no detailed analysis of failure/misalignment modes.
5. Interpretability and Qualitative Assessment: Figure 1 provides an intuitive visualization of GUARD’s impact on token probabilities and membership signals, and Figure 2 shows qualitative response evaluations. However, the paper would benefit from a more thorough qualitative error analysis in Sections 4.3 and Appendix A.7.6. For instance, is there a consistent type of answer (e.g., factual recall, definition, paraphrase) for which GUARD consistently underperforms or fails subtly compared to the fully fine-tuned model? The two examples in Figure 2 are not enough to uncover systematic trade-offs.
6. Ablation Study Limitations: The ablation study in Table 8 (Appendix A.3) presents only a simplified variation (“reordering only”) compared to full GUARD. More granular ablations—e.g., gold token anchoring only, diverse ranking strategies among non-gold tokens, variable top-$K$ thresholds, or temperature scaling effects—would offer sharper insights into which elements contribute most significantly to defense or utility.
7. Potential Exploitation of Non-Gold Distribution Rank: In Appendix A.2 the authors acknowledge the possibility that non-gold token ranking overlap between fine-tuned and pre-trained models may itself provide a membership leakage channel (possibly for an adaptive adversary). The presented evidence is that such overlap patterns vary randomly and the signal is weaker than the gold token, but this remains an empirical claim. Additional experiments (e.g., adversarial MIA variants that specifically test overlap exploitation) or further theoretical justification would strengthen the defense story.
8. Clarity in Naming and Consistency: Some notation and dataset/model naming is inconsistent or unclear. For example, Table 4 uses column headers like “Acuis” and “Gidush,” which seem like typos or confused references (likely “Arxiv” and “Github”). Such errors, though minor, can impede reproducibility and undermine confidence.

Required Figure and Table Engagement:
1. Referencing Figure 1 (Page 2): This figure is central; it visually illustrates the core premise that fine-tuning inflates the gold token probability (observable through the sharply heightened yellow bar) and therefore exposes the model to MIAs. GUARD by contrast restores the gold bar to its pre-trained level while preserving structure in the alternatives—a direct graphical argument for why attack success is neutralized. Furthermore, the radar plot (Figure 1d) concretely demonstrates GUARD’s strongly balanced performance across privacy, utility, scalability, and efficiency axes compared to other approaches, charting a clear visual superiority.
2. Referencing Table 2 and Table 6 (Pages 7 and 9): Table 2 shows AUC-ROC scores across datasets/attacks, consistently demonstrating GUARD’s ability to drive MIA success to near-random (≈0.5). Table 6, meanwhile, provides utility scores (GPT-4o and ROUGE-L), which—while slightly lower than full fine-tuning—still approach or surpass the utility of the best prior defenses, giving a quantitative measure of the privacy-utility trade-off.
3. Referencing Algorithm 1 and Equation in Section 3.2 (Page 5): This is where the technical heart of GUARD is laid out. While the loss is clearly defined, the implementation specifics (especially for top-$K$ retention, regularization strength $\lambda$, and distribution normalization under extreme class imbalance) warrant greater transparency for robust adoption.

**Questions:**

1. Can the authors clarify how the choice of top-$K$ for non-gold token retention impacts both the privacy leakage and model utility trade-off, especially for very large vocabularies ($>150$K tokens)? Would retaining more (or fewer) alternatives shift the defense-utility balance, and are there diminishing returns?
2. In Algorithm 1 and the main loss formulation, how sensitive is GUARD to the gold anchoring weight $\lambda$? Is there empirical/theoretical justification for the chosen default, and what guidance would you offer practitioners to tune this without heavy validation?
3. Have the authors considered or experimented with adaptive MIAs that attempt to exploit top-$K$ overlap or distributional rank instead of just the gold token? If so, please report findings and/or elaborate on how GUARD would defend against such adversaries.
4. Appendix A.2 gives some evidence that overlap-based signals are weak, but could you provide additional experiments (either adaptive MIAs or ablation studies) to confirm these results are robust to dataset/model scale and do not emerge as a secondary avenue for membership leakage?
5. Can the authors extend the ablation study to more granular variants (e.g., gold-only anchoring, alternative labeling of non-gold ranking, variable reweighting schemes, or temperature scaling effects) to better isolate the critical ingredients of GUARD’s empirical success?

---

> ### Comment · Area_Chair_Dy56 · 2025-11-21
> **Clarifying on LLM usage**
>
> Dear reviewer zLp7,
>
> Could you clarify on the LLM usage in the review? Particularly, we are concerned about evidence like  “Acuis” and “Gidush” in your review.

---

> ### Author Response · Authors · 2025-11-22
> **Response to Reviewer zLp7**
>
> Response to Question 1: Thanks for your question. As clarified initially in Appendix A.4, we set K = 1000 in the GUARD framework. This choice retains about 80% of the teacher’s probability mass, which effectively captures most useful dark knowledge to preserve utility (See Appendix A.6), while providing strong privacy, as shown by the main evaluation results (See Tables 2, 3, 4, 5). Importantly, prior work [1] has shown that even a much smaller setting (e.g., K = 32 using the Qwen-4B model) preserves most distillation benefits and achieves robust privacy, supporting the notion that the top logits dominate the learning signal. Therefore, K = 1000 strikes a more favorable balance that maximizes utility while maintaining strong privacy.
> [1]. BiLD: Bi-directional Logits Difference Loss for Large Language Model Distillation
>
> Response to Question 2: Thank you for the insightful question. In our experiments, we evaluated λ values from 0.1 to 0.5 across all model sizes (see table below). We found that λ = 0.5 consistently provides the best trade-off between stability and performance, effectively preventing excessive increases in gold-token probabilities while preserving utility across models. Consequently, we adopt λ = 0.5 as the default setting in all reported experiments. These results have been added to Appendix A.5 (Tables 10 and 11), we highlight it into blue color.
>
> **Evaluation of GUARD against multiple MIAs on LLaMA-3B across datasets under varying gold-weight \(\lambda\). Lower AUC-ROC (\(\downarrow\)) indicates stronger defense.**
>
> | MIA | PileCC (λ=0.1) | PileCC (λ=0.3) | PileCC (λ=0.5) | Wiki (λ=0.1) | Wiki (λ=0.3) | Wiki (λ=0.5) | HackerNews (λ=0.1) | HackerNews (λ=0.3) | HackerNews (λ=0.5) |
> |---|---:|---:|---:|---:|---:|---:|---:|---:|---:|
> | Zlib | 0.491 | 0.487 | **0.487** | 0.492 | 0.487 | **0.485** | 0.491 | 0.487 | **0.486** |
> | Loss | 0.616 | 0.536 | **0.497** | 0.618 | 0.546 | **0.498** | 0.615 | 0.526 | **0.501** |
> | Lowercase | 0.602 | 0.556 | **0.501** | 0.599 | 0.554 | **0.498** | 0.608 | 0.558 | **0.501** |
> | Mink | 0.708 | 0.579 | **0.504** | 0.701 | 0.582 | **0.502** | 0.702 | 0.575 | **0.501** |
> | Mink++ | 0.709 | 0.567 | **0.503** | 0.702 | 0.562 | **0.500** | 0.703 | 0.564 | **0.502** |
> | ReCall | 0.687 | 0.566 | **0.504** | 0.679 | 0.568 | **0.503** | 0.685 | 0.561 | **0.505** |
> | Con-ReCall | 0.670 | 0.580 | **0.505** | 0.668 | 0.578 | **0.502** | 0.667 | 0.576 | **0.500** |
> | Ratio | 0.657 | 0.547 | **0.499** | 0.654 | 0.543 | **0.503** | 0.650 | 0.544 | **0.502** |
> | Self-prompt | 0.722 | 0.596 | **0.516** | 0.713 | 0.598 | **0.510** | 0.719 | 0.588 | **0.509** |
>
> Response to Question 3: As discussed initially in Appendix A.2, we have studied this concern by examining top-50 token overlap on non-member data across PileCC and Wikipedia. Our analysis shows that fine-tuned (FT) and pre-trained (PT) distributions diverge only modestly, with an average increase in overlap of just 4~7% (Table 7, also attached below), substantially smaller than the up to 60% rise in gold-token probability induced by fine-tuning. Additionally, the overlap varies unpredictably across samples, indicating that top-K overlap is neither stable nor sufficiently discriminative to support reliable MIA attacks. Therefore, GUARD remains effective, as it explicitly regularizes gold-token anchoring, and adaptive MIAs that leverage distributional shifts in top-ranked tokens fail to consistently exploit meaningful leakage.
>
> **Comparison of FT vs. PT output distributions on non-member data.**
> Top-50 overlap is the percentage overlap between the top-50 predicted tokens of a fine-tuned (FT) model and its pre-trained (PT) counterpart (higher = more similar).
>
> | Model (FT vs PT) | PileCC-10k Top-50 overlap | PileCC-50k Top-50 overlap | Wiki-10k Top-50 overlap | Wiki-50k Top-50 overlap |
> |---|---:|---:|---:|---:|
> | GPT-Neo | 84.97 (+7.12) | 86.57 (+4.08) | 84.18 (+6.59) | 85.77 (+4.12) |
> | Qwen | 79.54 (+7.45) | 83.88 (+3.49) | 79.45 (+6.17) | 85.14 (+5.26) |
>
> Response to Question 4: Please refer to the table above. Our experiments confirm that these observations are consistent across different datasets and model scales, and do not introduce an alternative pathway for membership leakage.

---

> > ### Author Response · Authors · 2025-11-22
> > **Response to Reviewer zLp7**
> >
> > Response to Question 5: We adopt standard logit-level knowledge distillation (KD) and only modify the probability values afterward while explicitly preserving the teacher’s non-gold ranking. As such, ablations involving alternative non-gold labeling/ranking, additional reweighting schemes, or temperature-scaling sweeps are not meaningful for isolating GUARD. Modifying these components would alter the KD objectives themselves rather than evaluate the specific contributions of GUARD.
> > To isolate the impact of gold-token anchoring from standard knowledge distillation, we conducted an ablation comparing pure logit KD against GUARD for MIA defense. The results (attached below), now added in Table 8, demonstrate the effectiveness of GUARD beyond vanilla distillation.
> >
> > **Evaluation of GUARD against multiple MIAs under logit distillation using Qwen-3B as the teacher on diverse datasets (Qwen-1.5B as the student). Results are AUC-ROC; lower is better (↓).**
> >
> > | MIA | PileCC KD | PileCC **GUARD** | Wiki KD | Wiki **GUARD** | HackerNews KD | HackerNews **GUARD** | PubMed KD | PubMed **GUARD** | Arxiv KD | Arxiv **GUARD** | Github KD | Github **GUARD** |
> > |---|---:|---:|---:|---:|---:|---:|---:|---:|---:|---:|---:|---:|
> > | Zlib | 0.483 | 0.485 | 0.484 | 0.485 | 0.483 | 0.485 | 0.483 | 0.484 | 0.484 | 0.485 | 0.483 | 0.485 |
> > | Loss | 0.856 | 0.502 | 0.858 | 0.500 | 0.856 | 0.501 | 0.855 | 0.502 | 0.867 | 0.500 | 0.854 | 0.501 |
> > | Lowercase | 0.807 | 0.491 | 0.812 | 0.498 | 0.810 | 0.498 | 0.808 | 0.499 | 0.795 | 0.492 | 0.804 | 0.495 |
> > | Mink | 0.785 | 0.497 | 0.784 | 0.498 | 0.790 | 0.499 | 0.786 | 0.495 | 0.782 | 0.498 | 0.780 | 0.499 |
> > | Mink++ | 0.760 | 0.496 | 0.762 | 0.496 | 0.758 | 0.496 | 0.755 | 0.494 | 0.759 | 0.496 | 0.757 | 0.497 |
> > | ReCall | 0.822 | 0.501 | 0.824 | 0.501 | 0.821 | 0.498 | 0.822 | 0.496 | 0.819 | 0.497 | 0.823 | 0.499 |
> > | Con-ReCall | 0.834 | 0.499 | 0.826 | 0.502 | 0.828 | 0.499 | 0.835 | 0.498 | 0.829 | 0.499 | 0.832 | 0.501 |
> > | Ratio | 0.855 | 0.508 | 0.847 | 0.511 | 0.856 | 0.510 | 0.854 | 0.514 | 0.857 | 0.512 | 0.856 | 0.511 |
> > | Self-prompt | 0.913 | 0.511 | 0.912 | 0.514 | 0.910 | 0.512 | 0.912 | 0.512 | 0.899 | 0.513 | 0.908 | 0.512 |

---

> > > ### Author Response · Authors · 2025-11-22
> > > **Response to Reviewer zLp7**
> > >
> > > We are happy to address any additional concerns you may have and would sincerely appreciate your reconsideration of the score if you find our responses satisfactory.

---

> > > > ### Comment · Reviewer_zLp7 · 2025-11-26
> > > >
> > > > Thank you for the detailed responses. I appreciate the clarifications, and I will consider adjusting my score to 6.

---

### Author Response · Authors · 2025-12-01
**Responses summary for AC**

Dear AC: Thank you sincerely for overseeing the evaluation of our submission under the unusual situation involving a reviewer leak and reassignment. We appreciate your efforts to ensure a fair process.

Our work introduces **GUARD**, a *privacy-preserving framework* specifically designed to address the growing vulnerability of LLMs to **membership inference attacks (MIAs)**. Existing defenses often trade privacy for substantial performance losses. In contrast, GUARD leverages modified logit distributions that **retain the gold-token probability while anchoring the ranking of non-gold tokens**, enabling effective dark-knowledge transfer. This yields **a distilled student model with significantly enhanced privacy and minimal impact on utility.**

During the review phase, we provided additional ablations and clarifications that directly addressed key concerns:

- **Effectiveness beyond pure knowledge distillation (KD) (D4Vb Q1, zLp7 Q5).** Both reviewers questioned whether our gains were due to GUARD or to standard KD. New experiments (Appendix A3) show that **pure KD cannot offer GUARD’s privacy–utility trade-off**. Reviewers acknowledged this and raised their scores or mentioned raising scores (**D4Vb**).

- **Confusions with existing KD algorithms (7jM7 Q4).** We clarified that GUARD is explicitly designed as a privacy defense, not just another KD variant: it preserves the gold-token probabilities specifically to limit membership leakage.

- **Potential vulnerabilities and attacks (7jM7 Q3).** As discussed in Appendix A2, the risk raised by Reviewer 7jM7 does not materialize under our setting; the hypothesized vulnerability does not occur in our analysis and experiments.

- **Baselines, models, and resources (3UV9 Q1, 7jM7 w1 Q1).** We compare against all relevant public baselines we are aware of and evaluate across common backbone families (e.g., LLaMA-style, Qwen, GPT-Neo), within realistic GPU constraints. All experiments are aligned with prior MIA-related work. This provides coverage of widely used models rather than a single special case.

**Several weaknesses raised by reviewers stem from misunderstandings of our manuscript or are irrelevant. We briefly clarify them here:**

- **Scope of models / attacks / defenses (zLp7 w1, w2, w3; 3UV9 w1, w5; 7jM7 w1).** Our experiments are explicitly centered on **membership inference attacks**, although reviewer 3UV9 still asked: *“How does GUARD perform against specific privacy attacks, such as membership inference attacks (MIAs) or data extraction attacks?”* We evaluate **9 standard MIAs** following SOFT and recent LLM-MIA work, across several LLM families (GPT-Neo, Qwen, LLaMA-style) and multiple datasets (e.g., PileCC, Wiki). We also compare against **all relevant, reproducible MIA defense baselines** (e.g., unlearning, DP-batch training).

- **Implementation and practical details (3UV9 Q2, Q3).** The original manuscript and appendix provide **full implementation details** (Section 4.1, Appendix A6), while reviewer 3UV9 asked Q3: *“Could you provide more detailed implementation guidelines for GUARD, including any specific hyperparameters or tuning required for different models and datasets?”* We have added **clearer pointers from the main text** to these sections.

- **Positioning vs. the latest MIA work (zLp7 w1).** We **extended the related-work discussion** to explicitly cover recent MIA/LLM privacy frameworks and clarified that **GUARD is a plug-in training-time defense** that can be evaluated within those frameworks rather than a competing taxonomy.

- **Theoretical analysis and counterexample (7jM7 w2 Q2).** We clarified the assumptions of our theorem and showed that the reviewer’s counterexample **violates these assumptions**, so it does not contradict our result. **The theorem remains correct as stated**, and we improved the exposition for clarity.

- **Naming / “Acuis” and “Gidush” (zLp7 w8).** We re-checked our submission: **Table 4 uses “Arxiv” and “Github”.** The strings “Acuis” and “Gidush” do not appear in our paper. The previous AC raised the question of whether the reviewer used AI during the review.

Overall, **our work makes a novel and technically rigorous contribution**, with extensive and solid empirical results already demonstrated in the original submission. The additional clarifications and experiments provided during the review process **further reinforce both its soundness and significance**. We respectfully appreciate the AC’s consideration during the meta-review and hope **our work’s impact and value to the ICLR community will be recognized.**

---

### Meta-Review · Area_Chair_n49f · 2025-12-21

**Summary:**

This paper proposes a defense against Membership Inference Attacks (MIA) in an LLM fine-tuning setting. The key idea is, during fine-tuning, to preserve the probability assigned to the gold token while rearranging the probabilities of the remaining tokens to match the ordering by a regularly fine-tuned model. The method is evaluated over multiple datasets, backbone models, and MIA techniques, and the results show the proposed defense substantially reduces MIA accuracy.

After carefully reviewing the paper, reviews, and the rebuttal, I think the requested ablations and comparisons with additional baselines (distillation-based) have been adequately addressed. There is a concern about insufficient experimental coverage; however, the paper already considers a diverse set of datasets, backbone models, and MIA methods, so this concern is not well founded.

However, I do have several significant concerns based on a closer reading.

First, the evaluation of model utility is notably weak, despite extensive experiments on defense effectiveness, e.g., two datasets and two backbone models. More importantly, the evaluation data is synthetically generated using GPT-4o, and utility metrics are ROUGE-L and an LLM-as-a-judge, which are unreliable. This is because, although the paper assumes a fine-tuning scenario, it fine-tunes on pre-training data, and does not match them with real, existing downstream tasks with reliable test data and metrics. I recommend either (1) finding matching downstream tasks for each pre-trained data, e.g., SQuAD and other Wiki-related downstream tasks for Wikipedia, PubMedQA or other medical related tasks for PubMed, or coding tasks for Github; or (2) instead of fine-tuning on pre-training data, taking a train set of a downstream task, and use its test set and official eval metrics for utility evaluation.

Second, the overall experimental setup used an unrealistic epoch number, which is very critical in the defense-utility tradeoffs. The paper fine-tunes all models for 10 epochs, which is uncommon in practical fine-tuning scenarios, and would artificially significantly increase the MIA vulnerability (this likely explains the very high MIA accuracy observed for undefended models). In practice, developers would typically fine-tune for fewer epochs that maintain utility, which would naturally reduce MIA risk. In fact, simply reducing the epoch would be a competitive baseline defense (choosing the epoch in a way that it minimizes MIA risk while maintaining utility). Given that the number of training epochs is well-known to be a critical factor in MIA, not having discussion or ablations on it is a critical weakness of the work, and it remains unclear if reducing the epochs would flip the conclusion of the work.

Note to authors: I understand that these concerns were not raised in the original reviews. However, it is the AC’s responsibility to consider and highlight issues missed by the reviewers. Under normal circumstances, the AC would raise these points with the reviewers and engage in a joint discussion. This year, the AC is unable to discuss with reviewers and therefore must make a decision based solely on their judgement.

**Reviewer Concerns:**

(Noted in the summary)

**Reviewer Scores:**

(Noted in the summary)

---

### Decision · Program_Chairs · 2026-01-26

Reject